

# Constraints on Alpine Fault (New Zealand) Mylonitization Temperatures and Geothermal Gradient from Ti-in-quartz Thermobarometry

Steven Kidder[1], Virginia Toy[2], Dave Prior[2], Tim Little[3], Colin MacRae[4]

[1]Department of Earth and Atmospheric Science, City College New York, New York, 10031, USA
[2]Department of Geology, University of Otago, Dunedin, New Zealand
[3]School of Geography, Environment and Earth Sciences, Victoria University of Wellington, Wellington, New Zealand
[4]CSIRO Mineral Resources, Microbeam Laboratory, Private Bag 10, 3169 Clayton South, Victoria, Australia

*Correspondence to*: Steven B. Kidder (skidder@ccny.cuny.edu)

**Abstract.** We constrain the thermal state of the central Alpine Fault using approximately 750 Ti-in-quartz SIMS analyses from a suite of variably deformed mylonites. Ti-in-quartz concentrations span more than an order of magnitude from 0.24 to ~5 ppm, suggesting recrystallization of quartz over a 300° range in temperature. Most Ti-in-quartz concentrations in mylonites, protomylonites, and the Alpine Schist protolith are between 2 and 4 ppm and do not vary as a function of grain size or bulk rock composition. Analyses of 30 large, inferred-remnant quartz grains (>250 μm), as well as late, cross-cutting, chlorite-bearing

quartz veins also reveal restricted Ti concentrations of 2-4 ppm. These results indicate that the vast majority of Alpine Fault mylonitization occurred within a restricted zone of pressure-temperature conditions where 2-4 ppm Ti-in-quartz concentrations are stable. This constrains the deep geothermal gradient from the moho to about 8 km to a slope of 5 °/km. In contrast, the small grains (10-40 μm) in ultramylonites have lower Ti concentrations of 1-2 ppm, indicating a deviation from the deeper pressure-temperature trajectory during the latest phase of ductile deformation. These constraints suggest an abrupt, order of magnitude

change in the geothermal gradient to an average of about 60 °/km at depths shallower than about 8 km, i.e. within the seismogenic zone. Anomalously, the lowest-Ti quartz (0.24- 0.7 ppm) occurs away from the fault in protomylonites, suggesting that the outer fault zone experienced minor plastic deformation late in the exhumation history when more fault-proximal parts of the fault were deforming exclusively by brittle processes.

## 1. Introduction

The Alpine Fault is the major structure of the Pacific-Australian plate boundary through New Zealand's South Island. During dextral reverse fault slip, a <5 million year old, ~1 km thick mylonite zone (e.g. Sibson et al., 1979) was exhumed in the hanging-wall, providing a unique exposure of deep crustal material deformed to very high strains under boundary conditions constrained by present-day plate motions (e.g. Norris and Toy, 2014). The Alpine Fault is thus a rare location where the examination of plate boundary phenomena such as major earthquakes (Sutherland et al., 2007) or slow slip events (Chamberlain

et al., 2014) can be informed by observation of rocks that recently experienced these events. Because rock deformation was recent and continues today, aspects such as total strain and strain rate (Norris and Cooper, 2003), stress levels (Liu and Bird, 2002), exhumation rates (e.g. Little et al., 2005), and depth of seismicity (Leitner. et al., 2001) are known with a relatively high degree of certainty.

One poorly-constrained aspect of Alpine Fault deformation is the temperature distribution at depth along the structure. The geothermal gradient is a key factor in estimating rock strength and the position of the brittle-ductile transition (e.g. Kohlstedt et al., 1995), interpreting exhumation rates based on radiometric ages (e.g. Batt et al., 2000), and setting boundary conditions for thermomechanical models (e.g. Batt and Braun, 1999; Herman et al., 2007) and validating their results. Some constraints on



temperature are available for deep levels of the central Alpine Fault (e.g. Craw, 1997; Cross et al., 2015; Holm et al., 1989; Toy et al., 2010; Vry et al., 2004), however estimates of temperature vary significantly, disagreeing by as much as 200° at any given depth along the structure (Toy, 2007).

One tool with potential to improve understanding of the temperature distribution along the Alpine Fault is titanium-in-quartz thermobarometry (Thomas et al., 2010; Wark and Watson, 2006). In experiments where quartz is crystallized from a Si- and Ti-saturated aqueous fluid, trace concentrations of Ti-in-quartz are observed to be a function of both temperature and pressure (Thomas et al., 2010; Thomas et al., 2015). In both experimental and natural deformation, Ti concentrations in quartz are often reset during recrystallization (e.g. Behr and Platt, 2011; Kidder et al., 2013; Kohn and Northrup, 2009; Nachlas and Hirth, 2015;
Nachlas et al., 2014). While quantifying temperatures using Ti-in-quartz thermobarometry is complicated by uncertainty in Ti activity (Bestmann and Pennacchioni, 2015; Grujic et al., 2011; Nevitt et al., 2017) and differing calibrations (Huang and Audétat, 2012; Thomas et al., 2015), we find that variations in Ti-in-quartz concentrations in Alpine Fault rocks span a range large enough that these issues can be largely bypassed using independent constraints on maximum and minimum temperatures associated with quartz recrystallization. We have thus generated an extensive Ti-in-quartz data set that facilitates an independent
estimate of Ti activity and places new constraints on the temperatures and pressures at which mylonitization occur at deep levels of the fault.

## 2. Background

### 2.1 Geologic Setting

The Alpine Fault is a major transpressional structure extending ~400 km along the western slopes of New Zealand's Southern
Alps, connecting northeastward through the Marlborough fault zone to the Hikurangi subduction zone, and to the southwest to the Puysegur subduction zone (Sutherland et al., 2000). Modern day Alpine Fault systematics began with the onset of a transpressive phase of motion at ~5-8 Ma along the Australia-Pacific plate boundary (e.g. Batt et al., 2004; Cande and Stock, 2004; Walcott, 1998). Mylonites formed during this period have been exhumed in the hanging wall of the Alpine Fault (Norris and Cooper, 2007). The exposed mylonites are developed in the hanging-wall Alpine Schist (referred to below as "the schist"), a
complex, predominantly quartzofeldspathic assemblage also containing amphibolite and metachert. Age dating of the Alpine Schist suggests its protolith is an amalgamation of rocks formed at various times from the Carboniferous to Cretaceous (Cooper and Ireland, 2015). Barrovian metamorphic isograds in the non-mylonitic Alpine Schists that are the chief protolith of the Neogene-aged mylonites likely date to the late Cretaceous (e.g. Vry et al., 2004).

As the Alpine fault is approached from the east, schist fabrics are overprinted by a classic sequence of fault rocks: protomylonites, mylonites, ultramylonites, and finally cataclasites (Sibson et al., 1979, 1981). Most of the rocks analyzed for this study are from Gaunt Creek, the site of the first drill holes for the 2011 Deep Fault Drilling Project (DFDP, e.g. Boulton et al., 2014). One sample was analyzed from Stony Creek (Figure 1), 11 km southwest of Gaunt Creek.

### 2.2 Deformation History

Holcombe and Little (2001); Little et al. (2002a; 2002b) and Little (2004) provide detailed microstructural descriptions of the transition from the Alpine Schist to the Alpine Fault mylonite zone. In particular, Little et al. (2002a) recognized four main deformation episodes in the schist ($D_1$ to $D_4$). $D_4$ is Cenozoic deformation related to the evolution of the present dextrally convergent plate boundary (Neogene oblique collision between the Australian and Pacific Plates) and generally reinforces earlier





Mesozoic $D_3$ fabrics. The present schist foliation outside of the Alpine mylonite zone is mostly $S_{3–4}$. $D_4$ includes the high-strain mylonitic fabrics within ~1 km of the fault as well as a less conspicuous fabric resulting from distributed transpressional strain in more distal rocks. Neogene deformation incurred a vertical thickening and clockwise rotation (when viewed from the SW) and dextral shearing of the schists. The mylonitic $D_4$ imprint is only perceptible within 5 km of the Alpine Fault, where it is expressed

by a contractional finite shortening strain (post garnet growth) of <50% in a direction perpendicular to the foliation (Holm et al., 1989; Little et al., 2002a). The mylonitic foliation ($S_m$) that is clearly a product of Alpine Fault-related ductile shearing is subparallel to and strengthens these earlier fabrics, but results from higher strain and thus is aligned nearly parallel to the SE-dipping Alpine Fault.

Mylonitization began at depths of up to 35 km (Grapes, 1995; Little et al., 2005; Stern et al., 2007; Sutherland et al., 2007; Vry et al., 2004). Simple shear strains are >150 in the ultramylonites, as high as 120 in the mylonites, and 12-22 in the protomylonites. Characteristics such as boudinaged quartzose layers parallel to the shear zone, and reorientation of inherited, pre-Neogene lineations indicate that mylonitization involved a ductile flattening/shortening perpendicular to the fault with a stretch <1/3 (Gillam et al., 2013; Norris and Cooper, 2003; Toy et al., 2013). Strain rates were rapid and relatively well constrained at

roughly $10^{-12}$ s$^{-1}$ in the mylonite and ultramylonite zones and around $10^{-13}$ s$^{-1}$ in the protomylonite zone based on a range of geological and plate tectonic constraints (Norris and Cooper, 2003).

**2.3 Previous Temperature Constraints on Alpine Fault Deformation**

Thermal models indicate that rapid rates of exhumation in the hanging wall of the central Alpine Fault (~10 mm/yr; Norris and Cooper, 2007) must significantly raise the geothermal gradient in the vicinity of the fault, especially near the surface (e.g. Koons,

1987). Direct measurements of the geotherm in the upper 100's of meters of the crust near our study area are high (95 °/km, Shi et al., 1996; 125 °/km, Sutherland et al., 2017; 63°/km, Sutherland et al., 2012). Fluid inclusion evidence (e.g. Craw, 1997; Toy et al., 2010), the restriction of most earthquakes to depths <10 km (Boese et al., 2012; Bourguignon et al., 2015; Leitner. et al., 2001), and an observed decrease in temperature gradient near the base of the DFDP-2 borehole (Sutherland et al., 2017) indicate that the geothermal gradient is high near the surface but much lower at deeper levels (Niemeijer et al., 2016). Toy et al. (2008)

and Cross et al. (2015) suggested low geotherms in the ductile regime of 10°/km and 5°/km, respectively.

Thermobarometric data for the Alpine Fault mylonites and Alpine Schists (Beyssac et al., 2016; Cooper, 1980; Grapes and Watanabe, 1992; Grapes, 1995; Vry et al., 2004) indicate that mylonitization began at about 600 °C and 11 kbar. Toy et al. (2008) and Little et al. (2016; 2015) observed crystallographic preferred orientations in quartz that generally form at upper

greenschist to amphibolite facies conditions (~500-600 °C). Cross et al. (2015) estimated deformation temperatures of quartz in two samples of at least 450-500 °C based on the c-axis opening angle thermometer (Kruhl, 1996; Law, 2014) and microstructural evidence of subgrain rotation and grain boundary migration recrystallization in quartz (see also Little et al., 2015). Toy et al. (2010; 2008) noted the occasional presence of chlorite (generally after biotite) and green, low-Ti biotite, indicating some deformation at temperatures as low as greenschist facies in protomylonites and mylonites, with changes in biotite chemistry

indicating a change in temperature during mylonitic deformation of roughly −340 °C. The amphibolite-facies metamorphic assemblage associated with mylonitization is fairly consistent in diverse rock types regardless of distance to the Alpine Fault. Chlorite is not common in either the mylonites or ultramylonites, except in late-stage veins, extensional (C') shear bands, and footwall-derived ultramylonites (Toy et al., 2015).



## 3. Methods

Samples were analyzed at the California Institute of Technology, with a Cameca 7f Secondary Ion Mass Spectrometer (SIMS) using a $^{16}O^{-}$ primary ion beam. We used a beam current of 10-15 nA, a mass resolving power of 4000, field aperture of 100 mm, and analyzed masses $^{27}Al$, $^{30}Si$, $^{44}Ca$, $^{47}Ti$, $^{49}Ti$, and in some samples $^{56}Fe$. Prior to each analysis, we rastered a 50 x 50 mm area

for 60 s. Effective spot size for the analyses was about 10 μm. We used a regression line constrained though the origin to calculate Ti concentrations using National Institute of Standards (NIST) glasses 610 and 612 (434 ± 15 and 44 ± 5 ppm TiO2 respectively; Jochum et al., 2005). To account for matrix effects between quartz and NIST glass, a correction factor of 0.67, as determined by Behr et al. (2010) and independently tested by Kidder et al. (2013), was applied for the Ti data. As a Ti-blank, we used Herkimer 'Diamond', a natural quartz containing 4–5 ppb Ti (Kidder et al., 2013). Analyses of this natural blank suggest an

effective detection limit in this study of 78 ± 27 ppb. No blank correction was made since this value is far below Ti concentrations measured in the study.

For the subset of samples where Fe was measured (N=170), nearly all of the analyses with Ti concentrations >5 ppm also have high Fe/Si ratios ($^{56}Fe/^{30}Si > 0.007$, Figure 2). Analyses with Ti >5 ppm were also not observed to follow any discernable

patterns in CL intensity or position in the samples, e.g. nearer cores or rims of grains, while analyses with Ti <5 ppm do show such patterns in many cases. We interpret that all, or nearly all, of the analyses with Ti >5 ppm were contaminated, possibly by trace amounts of Fe-Ti oxides. Based on these observations, all analyses with Ti concentrations >5 ppm or $^{56}Fe/^{30}Si > 0.007$ were removed from the dataset. We were unable to identify any clear criteria to similarly filter the data using Al or Ca concentrations.

Cathodoluminescence (CL) images were acquired prior to SIMS analyses on most of the areas where Ti was analyzed using the Zeiss variable-pressure field-emission scanning electron microscope (SEM) at the University of Otago. Images were collected using a variable-pressure secondary electron (VPSE) detector operated at high vacuum, 30 kV accelerating voltage and 7 nA beam current. This detector is sensitive in the range 300–650 nm. Hyperspectral CL maps (MacRae et al., 2013) were acquired in one sample using a JEOL 8500F electron microprobe equipped with an ocean optics QEPro spectrometer tuned to collect from

200-960nm. The sample for hyperspectral CL mapping was polished with colloidal silica, coated with 15nm of carbon and analyzed at 20kV accelerating voltage, 30nA beam current, dwell time of 40 ms per pixel, and a defocused beam and pixel size of 2 μm. Hyperspectral CL maps were displayed and processed using in-house software, Chimage (Leeman et al., 2012). The hyperspectral maps were fitted at each pixel using a least squares approach with a set of three Gaussian distributions: the $Ti^{4+}$ peak at 470 nm (Leeman et al., 2012),the near infrared peak at 729 nm, and the non-bridging oxygen hole center at 646 nm

(Kalceff and Phillips, 1995).

Measurements of grain size, distance between analysis spots and features such as grain boundaries or Ti-bearing phases were carried using an optical microscope following SIMS analyses. In order to determine if rutile, a minor constituent in some of the rocks, is present, several hundred to several thousand bright grains in backscatter imagery were automatically characterized in

each sample using energy-dispersive X-ray spectroscopy (EDS) with the Oxford Instruments particle analysis "Feature" algorithm on the University of Otago SEM.



## 4. Description of samples

### 4.1 Schists and Protomylonites

Most of the quartz in the protomylonites and schists has a grain size of 20-200 μm, though grains as large as 2-4 mm are found. Larger grains commonly have patchy undulose extinction, contain subgrains, and are elongated (axial ratios from 1:2 to 1:10)

with a distinct shape preferred orientation (SPO). High angle boundaries between the large quartz grains are generally interlobate. In many samples, small (<50 μm) grains are prevalent adjacent to these interlobate boundaries. Feldspars form porphyroclasts and show little evidence of crystal plasticity--only weak undulose extinction and in some cases deformation twins. Since both subgrains and, more commonly, relatively large grains with interlobate high-angle grain boundaries are observed, the observed quartz microstructures resemble those of the regime 2/3 transition to lower regime 3 described by Hirth

and Tullis (1992) in experimentally deformed quartzites. C' shear bands, some containing a retrograde chlorite-bearing assemblage, are pervasively developed in the protomylonites (Gillam et al., 2013). These contain recrystallized quartz grains as small as 10-20 μm.

### 4.2 Mylonites and Ultramylonites

Towards the Alpine Fault the mineral phases in the quartzofeldspathic mylonites and ultramylonites become more mixed and

grain size (all grains, i.e. recrystallized plus inherited grains) decreases. S-C and C' fabrics are abundant. Minor chlorite is present in some rocks but is less abundant than in the protomylonites. Within a few hundred meters of the Alpine Fault, phase mixing is pronounced, particularly in ultramylonite layers, and domains of pure quartz are restricted to rare foliation-parallel veins and metachert layers. Quartz aggregates generally consist of slightly elongate (axial ratios mostly >2:1) grains of size 40 - 100 μm. Except in the metacherts and deformed pure quartz veins, large quartz grains with undulose extinction are much rarer in

the ultramylonites and mylonites than in the protomylonites. The maximum typical long axis of quartz grains in the mylonites is around 1 mm and decreases in size towards the fault. The larger quartz grains in the mylonites are bounded by interlobate high-angle boundaries, with only weak undulose extinction and occasional aggregates of subgrains. In mylonitic metachert samples, secondary phases are often encased in quartz, which often forms elongate, ribbon-like foliation-parallel grains bounded by micas. These quartz grains display castellate microstructures, pinning and other indicators of high-temperature grain boundary

migration (Jessel, 1987; Little et al., 2015). Within ~100 m of the fault, rare concentrations of quartz are often strongly recrystallized with a recrystallized grain size on the order of 10-20 μm. In these samples, ribbon grains are common and often filled with 10-20 μm subgrains (Fig. 3) suggesting subgrain rotation recrystallization was a dominant process (Hirth and Tullis, 1992; Stipp et al., 2002).

## 5. Cathodoluminescence

Cathodoluminescence ("CL") intensity generally correlates with Ti concentration such that bright CL corresponds with high Ti concentrations, and several studies have utilized panchromatic CL signal as a proxy for Ti concentrations (Bestmann and Pennacchioni, 2015; Nevitt et al., 2017; Spear et al., 2012). Typical CL images of the Alpine Fault mylonites and protomylonites show a homogeneous illumination (e.g. Figure 4a), generally with an apparent thin dark rim around the edges of grains. Rare samples contain grains with gradients in CL, typically large grains with higher CL that diminishes gradually towards grain edges.

In these cases, Ti concentrations are generally correlated with panchromatic CL, however several exceptions to this pattern were noted. The typical association of recrystallized grains with darker-CL and low Ti (e.g. Kidder et al., 2013) was rarely observed in the mylonites and protomylonites, but is common in the ultramylonites. The sample analyzed using hyperspectral CL has a



homogeneous Ti concentration, so the typical association of blue light intensity and higher Ti (Wark and Spear, 2005) was not observed, and instead variation in the blue wavelength signal is seen to be a function of grain orientation (Figure 4). We note that both these anomalous results and those of Kidder et al. (2013), who found a brighter blue CL signal in recrystallized areas, were observed in quartz with Ti concentrations much lower than measured by Wark and Spear (2005).

## 6. Ti-in-quartz concentrations: observations

### 6.1 General observations

The bulk composition of Alpine Schist samples does not appear to be a significant factor controlling the Ti concentration in their quartz grains. Neighboring felsic and mafic schists have ranges of Ti concentration that broadly overlap, although a few amphibolite and metachert samples have up to ~1 ppm higher Ti concentrations than nearby quartzofeldspathic schists (Figure 5). The presence of chlorite and rutile in samples (Table 1) is also not systematically associated with higher or lower Ti concentrations.

The range of Ti values increases with decreasing grain size (Figure 6)(c.f. Kidder et al., 2013). Grains larger than ~200 µm in the two creeks have indistinguishable Ti concentrations, but smaller grains in two samples from Stony Creek (including one from Cross et al., 2015) have lower Ti values than observed in Gaunt Creek (Figure 6). Figure 7 plots Ti concentration versus distance to grain boundaries (Figure 7a) and distance to any dark mineral likely to contain Ti as a major component (Figure 7b). Ti concentrations have values of 2-4 ppm Ti for nearly all analyses further than 50 µm from grain edges, whereas lower values gradually become more common closer to grain boundaries. A paucity of Ti values <1 ppm Ti at distances <5 µm may be a function of minor contamination of analyses that include grain boundaries (figure 7a). The potential effect of proximity to minerals containing non-trace amounts of Ti (mainly biotite and Fe-Ti oxide) appears to be minimal (Figure 7b). The lowest Ti quartz grains (Ti <0.8 ppm) were only found >50 µm from Ti-bearing phases (Figure 7b). In one metachert sample (77913), a transect away from a large rutile grain showed no variation in Ti with distance from the grain.

### 6.2 Observations: Schists, Protomylonites, and Mylonites

Because high temperature dynamic recrystallization can involve resetting of Ti contents as grain boundaries sweep through quartz (e.g. Grujic et al., 2011), it is non-trivial to ascertain whether the limited range of 2-4 ppm Ti in grains coarser than ~250 µm (Figure 6) indicates initial Ti concentrations prior to Alpine Fault motion, or whether they were reset by prolonged high temperature deformation. Most quartz-rich layers have relatively homogeneous Ti concentrations (Figure 4) as commonly observed in samples that experienced protracted high-temperature metamorphism (e.g. Spear and Wark, 2009).

In the Gaunt Creek mylonites and protomylonites, recrystallized grains of all sizes generally have Ti contents that are also in the same 2-4 ppm range as the coarser grains, although some moderate-sized (20-200 µm) grains with Ti concentrations in the range 1-2 ppm are also present (Fig. 8). In individual samples, Ti concentrations are often homogenous, varying with no systematic spatial pattern between about 2 and 3 ppm (e.g. Figure 4). Some samples do contain local, minor-but-systematic variations in Ti within this range of values, however no over-arching pattern was observed. For example, in sample 77886, Ti increases from ~2.5 to 3.5 ppm where a quartz layer is pinched, recrystallized, and finer-grained adjacent to a garnet porphyroclast. Quartz in a similar microstructural setting was not associated with a consistent change in Ti in the Alpine Fault rocks studied by Cross et. al. (2015). Two metachert samples (77920 and 77923) contain 400 µm thick quartz layers with low-Ti central areas in the range 1-2.5 ppm Ti that increase to ~3 ppm at their edges. A similar pattern is observed in protomylonite sample 77982, and could be



interpreted as indicating lower initial Ti values than 2-4 ppm for these rocks. Several large quartz grains in sample 77982 also have bright-CL cores suggesting different "initial" Ti in the metachert and quartzofeldspathic portions of the sample. At a larger scale, there is a gradual decrease in average Ti concentrations of 3-4 ppm at structural distances of 500 m from the Alpine Fault to values closer to 2-3 ppm at 1100 m (Figures 9). This pattern is particularly evident in coarse grains (Figure 9 inset).

Two samples, a protomylonite and a mylonite (77875 and 77925), contain late, chlorite-bearing quartz veins that cut foliation. Quartz in both veins shows evidence of minor dynamic recrystallization and has Ti concentrations of 2-4 ppm. Given the high shear strains experienced by the Alpine Fault mylonites (Norris and Cooper, 2003), these late-stage veins formed during the last few percent of penetrative mylonitic strain. The presence of chlorite in these veins constrains their emplacement temperatures and therefore the cessation of mylonitization to within the chlorite stability field, a relatively wide range of ~360-500 °C in Alpine Fault rocks (Beyssac et al., 2016; Vry et al., 2007).

Fine recrystallized quartz in two of three C' type shear bands shows evidence of lower Ti-in-quartz concentrations than surrounding quartz. A C' shear band from a Stony Creek sample (same as sample ST_12 in Little et al., 2016) shows a marked decrease in Ti concentrations within the shear band (Figure 10). In Gaunt Creek, quartz in one C' band from near the mylonite-protomylonite boundary (77886) shows a lower Ti content (~2.25 ppm) than quartz in the foliation that is cross-cut (~3.5 ppm Ti). A shear band in protomylonite sample 5.2.1.13, however, shows little to no decrease in Ti in a (chlorite-absent) C' band.

### 6.3 Observations: Ultramylonites

The population of quartz grains in ultramylonites of size >30 μm generally have indistinguishable, or slightly lower, Ti concentration when compared to rocks further up section (Figure 8). Finer grains, however, are distinctly shifted to lower values (1-2.5 ppm vs. values of 2-4 ppm in the mylonites and protomylonites). This trend is associated with a decrease in average quartz grain size and a deviation from the higher Ti concentrations in adjacent coarser grains (Figures 9). This pattern is evident in individual transects from coarser-grained areas to finer-recrystallized areas with decreased Ti concentrations of 1-2 ppm in both ultramylonite samples as well as the closest mylonite sample (77911) to the Fault (e.g. Figure 3).

## 7. Discussion

### 7.1    Evolution of Ti concentrations

### 7.1.1    Initial Ti-in-quartz concentrations were 2-4 ppm

Coarse polygonal quartz grains in non-mylonitic schist outside the present study area are inferred to have formed in the Mesozoic during an extended, non-orogenic period when the rocks resided at high temperature in the middle to lower crust (Little, 2004), i.e. prior to their Neogene exhumation resulting from shearing on the Alpine Fault. The roughly 30 coarse grains analyzed (>250 μm) from schist and mylonite samples in our study contain a restricted range of Ti concentrations of 2-4 ppm (Figure 6), and given their large size (some >1 mm), we also infer these to be Mesozoic grains. The characteristic diffusion timescale (e.g. Spear, 1995) for 250 μm grains using the experimental data of Cherniak et al. (2007) at a peak Alpine Fault mylonitization temperature of ~550°C (Cross et al., 2015; Toy et al., 2010) is 6 b.y., seemingly barring the possibility that these grains were reset by diffusional processes during mylonitization lasting a few million years in the Neogene. We interpret that the 2-4 ppm Ti-in-quartz concentrations in large grains were stabilized during Mesozoic metamorphism, prior to Neogene Alpine Fault related deformation.



The significance of the apparent trend in Ti concentrations of the largest grains in the Gaunt Creek section towards higher Ti concentrations at deeper structural levels (Figure 9 inset) is unclear. If the variation in initial Ti concentration were purely the result of differences in initial depth along a vertical column of rock, the position of the isopleths in Figure 11 would indicate a

stretch perpendicular to foliation of 1/7, more than twice the foliation-perpendicular component estimated in previous studies (Holcombe and Little, 2001; Holm et al., 1989; Toy et al., 2013). It is worth noting, however, that because of the extreme simple shear strains experienced by the Alpine Fault mylonites, the various hanging wall rocks sampled for this study and presently separated by distances <1 km, were originally separated by >100 km (Norris and Cooper, 2003). Given these large offsets, it is possible that the large apparent shortening was a function of pre-Alpine Fault regional variations in geothermal gradient (i.e. a

regional variability in initial Ti concentration at a given depth), and/or shallowing of the hypothesized deep, semi-horizontal continuation of the Alpine Fault towards the east (Batt and Braun, 1997; Cox and Sutherland, 2007; Little et al., 2002a).

### 7.1.2    Protomylonite & Mylonite Deformation occurred along a Geothermal Gradient of 5 °/km

Two observations support a hypothesis that the pre-Alpine Fault 2-4 ppm range of Ti-in-quartz concentrations was stable for nearly the entire period of protomylonite and mylonite deformation. First, slightly-deformed chlorite-bearing quartz veins that

cross-cut foliation also have Ti concentrations of 2-4 ppm. Second, small quartz grains generated in the mylonites and protomylonites during Alpine Fault deformation contain similar concentrations of Ti as coarser grains (e.g. Figures. 4, 8). This is atypical behavior, since increasingly finer grains formed during exhumation generally show systematic changes in Ti concentration (e.g. Behr and Platt, 2011; Kidder et al., 2013; Kohn and Northrup, 2009; the Alpine Fault ultramylonites described in this manuscript). We suggest that, as commonly observed, gradually increasing stresses due to strengthening during

cooling also led to an increasingly fine-grained overprint during exhumation of the Alpine Fault mylonites. The unvarying Ti-in-quartz concentrations in the Alpine Fault can be explained then if the exhumation path happened to coincide with a contour of constant-Ti concentration in pressure-temperature space (an "isopleth," Figure 11). The isopleths plotted in figure 11 are based on the experimental results of Thomas et al. (2010) and correspond to a slope of 5 °/km, a plausible value that is actually identical to the deep geothermal gradient proposed by Cross et al. (2015) based on different criteria. Thus, despite a significant

decrease in temperature, Ti levels of 2-4 ppm remained stable during the bulk of Alpine Fault mylonitization.

### 7.1.3    Ultramylonite Deformation involved a decrease in stable Ti levels to 1-2 ppm

In the ultramylonites, Ti-in-quartz concentrations in grains coarser than 50 μm are generally indistinguishable from those in protomylonites and mylonites, however lower Ti concentrations (1-2 ppm) are much more common in fine recrystallized grains in the ultramylonites (Figures 8). We interpret the clustering of Ti-in-quartz values in the range 1-2 ppm in the recrystallized

areas of several ultramylonite samples as indicating the equilibrium concentration associated with the latest phase of quartz recrystallization near the Alpine Fault. Note that the spread of Ti-in-quartz concentrations in fine grains generated during late recrystallization (Figures 8, 9, 10) suggests that dynamic recrystallization was not 100% effective in shifting Ti concentrations to equilibrium values, as observed elsewhere (e.g. Bestmann and Pennacchioni, 2015; Haertle et al., 2013; Kidder et al., 2013; Nevitt et al., 2017).

**7.1.4    Latest Deformation Involving Quartz Recrystallization Occurred Off-Fault**

The lowest Ti concentrations measured in this study (0.5-0.8 ppm) come from within a chlorite-bearing C' shear band in a protomylonite sample from Stony Creek (Figure 10). C' shear bands are abundant in Alpine Fault protomylonites (Little et al.,

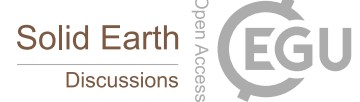



2016) and elsewhere, and are widely understood to be late deformation features (e.g. Toy et al., 2012) accommodating minor amounts of strain relative to the total mylonitic deformation (Gillam et al., 2013). The low Ti-in-quartz concentrations in this shear band, as well as the decreased Ti concentrations in one of two Gaunt Creek shear bands analyzed, are consistent with resetting during a late stage of low-temperature deformation.

The Stony Creek mylonite sample analyzed by Cross et al. (2015) contains very low Ti concentrations (0.2-0.4 ppm) in relatively large grains (up to 130 μm) not microstructurally associated with any obvious late deformation phase. Thus, at face value, the very last quartz grains to recrystallize were not in localized C' zones, but were instead fairly moderate-sized quartz grains not associated with shear bands or another type of microstructure indicative of late, low-temperature deformation. Anomalously low

Ti concentrations were also observed locally in Gaunt Creek, but the Ti values are not as low (0.5-1 ppm; Figure 6). It is unclear if these observations are influenced by some difference between Stony and Gaunt creeks, but it is noteworthy that the two samples from Stony Creek have the lowest Ti-in-quartz measurements observed. Possibly deformation in Stony Creek continued at lower temperatures than in Gaunt Creek (11 km to the NE) or there was a difference in the geothermal gradient between the two creeks. The two creeks show at least one compositional difference (more metacherts in Stony creek), so it is also possible

that some systematic compositional differences might have reduced Ti activity in Stony creek. If differences in bulk composition between the two creeks were the cause, however, we would not expect coarse grains in the two creeks to have the indistinguishable Ti concentrations seen in figure 6, nor would we expect an absence of a compositional effect locally in Gaunt Creek samples (figure5).

In any case, the presence of Ti concentrations <0.8 ppm along C' shear bands and elsewhere in the protomylonite zone, and absence of such values closer to the fault suggests that the latest phase of deformation involving dynamic recrystallization of quartz did not take place in the ultramylonites as might be expected based on general models of shear zones involving increased localization at shallower depths (e.g. Sibson, 1977). Instead, the latest phase of deformation involving quartz recrystallization occurred in a broad region of protomylonites outside the fault core. Toy (2007) reached the same conclusion based on

observation of a more retrograde mineral assemblage associated with C' shear bands. An explanation for this may be illustrated in strength-depth diagrams plotting flow laws at different strain rates (e.g. Kidder et al., 2012): near the brittle-ductile transition, deformation at slow strain rates will be in the plastic regime, while at the same pressure-temperature conditions, faster strain rates can be accommodated with less resistance by brittle mechanisms. We suggest that near the BDT, as strain rates increased in the narrow (<10 m-wide) cataclastic zone, deformation away from this localized zone remained slow enough to favor some

dislocation creep. Put differently, the brittle-plastic transition extended to deeper levels near the fault than away from it, as predicted in numerical models of the Alpine Fault by Ellis and Stöckhert (2004).

### 7.2 Ti Activity

The activity of Ti has proven to be a major source of uncertainty in previous studies (e.g. Grujic et al., 2011; Nevitt et al., 2017), apparently varying by nearly an order of magnitude in previous studies of quartzofeldspathic rocks from values as high as 1

(Kidder et al., 2013) to 0.2 (Grujic et al., 2011). The lack of knowledge of Ti activity and the factors that control it complicates quantitative estimation of temperature using the TitaniQ thermobarometer. The question of Ti activity is further hindered by competing calibrations of TitaniQ (Huang and Audétat, 2012; Thomas et al., 2010; Thomas et al., 2015). While we employ the calibration of Thomas et al. (2010; 2015), we note that using the Huang and Audetat (2012) calibration would not substantially

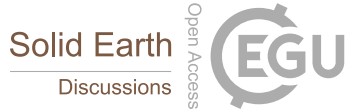

change our results (apart from the estimated Ti activity) because the two calibrations have similar slopes in pressure-temperature space at pressures >2 kbar (see Figure 6 of Kidder et al., 2013).

Despite these uncertainties, sufficient independent information is available for Alpine Fault rocks to constrain their Ti activity to
a very low value of 0.1 (using the Thomas et al., 2010 calibration). This value is estimated assuming that the lowest observed Alpine Fault Ti-in-quartz concentrations of ~0.2 ppm represent deformation at the low-temperature limit for dynamic recrystallization of quartz of 300 °C (Dresen et al., 1997; Dunlap et al., 1997; Stipp et al., 2002; Stöckhert et al., 1999; van Daalen et al., 1999), and using the relatively steep (>50 °/km) geothermal upper crustal geothermal gradient indicated by fluid inclusions (e.g. Toy et al., 2010)(figure 11) and borehole data  in the vicinity of the study area (Sutherland et al., 2017;
Sutherland et al., 2012). The lower limit of 0.2 ppm Ti has also been observed in Alpine Fault rocks to the south near the Haast River (Kidder et al., 2016a). Using a higher Ti activity would require that recrystallization of quartz took place at unrealistically low temperatures (figure 11). A Ti activity of 0.1 also places the estimated initial Ti concentration of ~3 ppm at a temperature of 600 °C near the base of the crust, consistent with the thermobarometric constraints from Vry et al. (2004) that mylonitization began at around 600 °C and 11 kbar. A similar analysis using the Ti-in-quartz calibration of Huang and Audetat (2012) would
involve a Ti activity in the range 0.5-0.8 (e.g. as adopted by Cross et al., 2015).

While it is convenient to assign a Ti activity to an entire suite of rocks such as our sample set, it may be the case that there are sample- or micro-scale variations in Ti activity due to differences in bulk or fluid composition. Ti activity could also vary with temperature (Ashley and Law, 2015). An absence of Ti values less than ~1 ppm at distances <50 μm from phases containing
stoichiometric quantities of Ti (Figure 7B) may indicate such variation. This pattern could be explained if Ti activity was locally higher than 0.1 in the vicinity of such minerals at low temperatures. At temperatures of 400-500° where we interpret the bulk of recrystallization happened, there is no indication of such elevated Ti activity (Figure 7B), however, a change in Ti activity to 0.5 in the immediate vicinity of Ti-bearing phases at temperatures of 300° would make Ti concentrations of 1 ppm stable at these conditions during the latest phases of quartz recrystallization, and explain the absence of lower Ti quartz in these areas.

**7.3    Geothermal Gradient of the Alpine Fault**

Considering that long term average strain rates are well-established in the Alpine Fault mylonites at ~$10^{-12}$ (Norris and Cooper, 2003), we can place a rough estimate on the minimum temperature of mylonite deformation using a piezometer and flow law. Although smaller grains are present (e.g. Figure 8), the average recrystallized grain size in the Alpine Fault mylonites is in the range of 40-70 μm (Lindroos, 2013; Little et al., 2015; Toy, 2007). This range corresponds with stresses of 31-44 MPa (Cross et
al., 2015) and a temperature of 430-480 °C using the quartz flow law from Kidder et al. (2016b). The intersection of the 3 ppm isotherm and this temperature range occurs at depths of 0-10 km (Figure 11), thereby requiring an order of magnitude change in slope from about 5 °/km to at least 60 °/km at depths of 8 km or less. We adopt the values (60 °/km above 8 km) that minimize the magnitude of the kink, although shallower kinks and larger upper crustal geothermal gradients would also fit the data presented here. This upper crustal geotherm falls close to the geothermal gradient of 63 ± 2 °/km measured in the upper 140
meters of the DFDP- 1B borehole site at Gaunt creek (Sutherland et al., 2012), although it is clear that there is considerable local variation in the upper crustal geothermal gradient likely due to fluid circulation associated with topography (Sutherland et al., 2017). The Alpine Fault geothermal gradient predicted here is very similar to that presented by Cross et al. (2015)—differing only in the slightly shallower position of the kink at 480° in this study (vs. 495°), and the correspondingly larger upper-crustal geotherm in our analysis (Figure 11). The position of the kink corresponds roughly to the maximum depth of earthquakes near





the study area (Boese et al., 2012; Leitner. et al., 2001), and could be explained by rapid cooling at depths where meteoric fluids are able to penetrate due to fracturing associated with earthquakes.

Despite the similarities between the geothermal gradient presented here and that of Cross et al. (2015), there are significant differences in our microstructural interpretations stemming from the much larger dataset and variety of rock types examined in this study. Cross et al. (2015) interpreted that the restricted range of Ti concentrations in their two protomylonite samples were established rapidly, in a narrow window of temperatures (450-500°) above the kink in the geotherm. Alternatively, we attribute the restricted range of Ti concentrations to deformation along a constant Ti isopleth below the kink—a scenario not considered by Cross et al. (2015). In their interpretation, the microstructures are essentially a snapshot of a rapidly changing grain configuration, whereas we hypothesize the partial preservation of Ti concentrations and individual larger grains from the entire time period including and predating Alpine Fault deformation. We propose the new hypothesis because of our identification of two phases of Ti-in-quartz behavior (e.g. Figure 12). In the earlier phase (phase 1), coarse grains with a homogenous Ti concentration were established (Figures 6, 8). The development of finer grains during late phase 1 deformation was not associated with significant systematic changes in Ti concentration (e.g. Figures 4, 8). Alternatively, phase 2, associated with ultramylonites (Figures 3, 8) and C' fabrics (Figure 10), is marked by decreased Ti-in-quartz concentrations in recrystallized grains as the exhumation path ceased following a constant-Ti isopleth and begin crossing Ti isopleths.

## 8. Conclusions

The vast majority of the mylonitic deformation of the Alpine Fault occurred while rocks were exhumed from depths of around 35 km to around 8 km while cooling by only 120° (from 600°-480°) along a geothermal gradient of 5°/km. Assuming a constant exhumation rate, cooling rates increased at a depth of 8 km by an order of magnitude as the geothermal gradient increased to about 60°/km. The transition at 8 km corresponds to the lower limit of seismicity and the formation of ultramylonites in the fault core. We infer that the change in geothermal gradient was facilitated by increased fluid circulation due to seismically-induced permeability changes. Deformation at temperatures <400° was dominated by brittle processes such as cataclasis on and near (within ~30 m of) the Alpine Fault, and, until a temperature of 300° was reached, minor ductile deformation in protomylonites away from the fault. Despite major uncertainties in Ti activity, Ti-in-quartz data can provide valuable quantitative constraints on deformation temperature and geothermal gradients.

## Acknowledgments

We are grateful for conversations with Andrew Cross and to Ashfaq Khan for compiling microstructural observations and data processing. Yunbin Guan provided helpful assistance with the SIMS, and Brent Pooley prepared thin sections. Funding for the project was supplied by NSF grants IRFP-1064805 and EAR-1524602.



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



**Table 1**. Information on individual samples. All samples are from Gaunt Creek with the exception of Stony Creek sample ST-12c_iv. GPS data use the New Zealand Transverse Mercator Projection 2000. Distances are estimated structural distances measured perpendicular to foliation. Rutile and Chlorite columns indicate whether these minerals were observed ("y") or not ("n"). Sample identified by numbers 77*** are archived in the Otago University collection.

| Sample | Northing | Easting | Distance (m) | Composition | Structure type | Rutile | Chlorite |
|--------|----------|---------|--------------|-------------|----------------|--------|----------|
| 77921 | 5,200,537 | 1,382,891 | 83 | qfs | mylonite | y | n |
| 77911 | 5,200,537 | 1,382,891 | 118 | qfs | mylonite | n | n |
| 77920 | 5,200,511 | 1,383,128 | 133 | mafic part | mylonite | y | trace |
| 77920 | 5,200,511 | 1,383,128 | 133 | chert part | mylonite | y | trace |
| 77821 | 5,200,508 | 1,382,904 | 140 | qfs/mafic | ultramylonite | y | y |
| 77913 | 5,200,511 | 1,382,888 | 137 | chert | mylonite | y | y |
| 77915 | 5,200,511 | 1,382,888 | 137 | mafic part | mylonite | y | y |
| 77915 | 5,200,511 | 1,382,888 | 137 | vein in mafic | mylonite | y | n |
| 77923 | 5,200,516 | 1,383,157 | 149 | mafic part | mylonite | y | y |
| 77923 | 5,200,516 | 1,383,157 | 149 | chert part | mylonite | y | n |
| 77912 | 5,200,551 | 1,382,908 | 159 | augen mafic feldspar | ultramylonite | n | y |
| 77925 | 5,200,564 | 1,383,283 | 246 | qfs | mylonite | y | n |
| 77925 | 5,200,564 | 1,383,283 | 246 | vein in qfs | mylonite | y | in late vein |
| 77886 | 5,200,429 | 1,383,283 | 292 | qfs | mylonite/proto | y | y |
| 77887 | 5,200,426 | 1,383,287 | 293 | mafic part | mylonite | n | n |
| 77887 | 5,200,426 | 1,383,287 | 293 | chert part | mylonite | y | y |
| 77900 | 5,200,409 | 1,383,570 | 461 | chert | schist | y | n |
| 5.2.1.13 | 5,200,409 | 1,383,570 | 461 | qfs | proto | n | n |
| 5.26.1.18 | 5,200,409 | 1,383,570 | 461 | qfs | proto | n | n |
| 77930 | 5,200,362 | 1,383,619 | 515 | mafic | proto | n | y |
| 77932 | 5,200,344 | 1,383,638 | 541 | mafic part | proto | y | n |
| 77932 | 5,200,344 | 1,383,638 | 541 | chert part | proto | y | n |
| 05.2.1.11 | 5,200,344 | 1,383,638 | 541 | qfs | proto | n | y |
| 05.2.1.11 | 5,200,344 | 1,383,638 | 541 | vein in qfs | proto | n | n |
| 77878 | 5,200,202 | 1,383,798 | 741 | qfs | proto/schist | y | y |
| ST-12c_iv | 5,193,839 | 1,374,804 | 800 | qfs | proto | n | y |
| 77875 | 5,200,152 | 1,383,943 | 869 | qfs | proto/schist | n | in vein |
| 77982 | 5,200,115 | 1,384,037 | 942 | qfs | schist | y | n |
| 77980 | 5,200,163 | 1,384,269 | 1090 | qfs | schist | y | y |





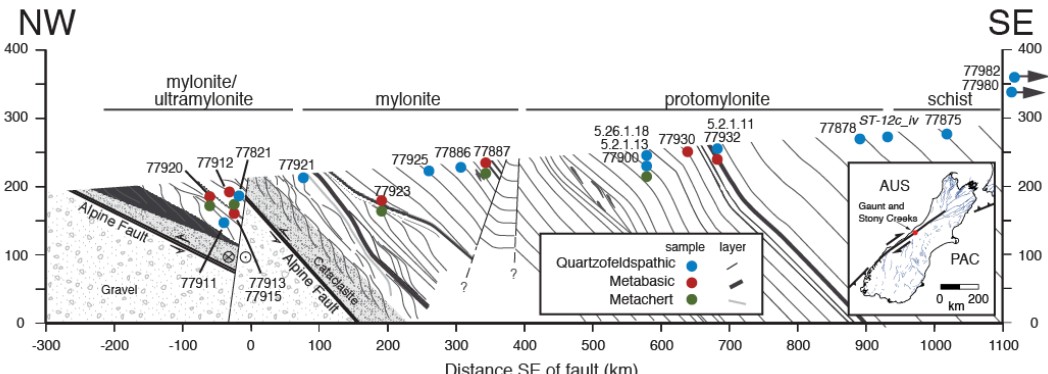

**Figure 1**. Schematic cross section of Gaunt Creek after Toy et al. (2008) showing approximate sample locations. Inset shows the
5    S. Island of New Zealand and the locations of Gaunt and Stony creeks. The position of the the Stony Creek sample (ST-12c_iv)
is projected into an approximate position based on its structural distance from the Alpine fault.

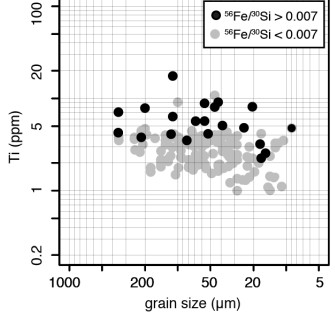

**Figure 2.** Plot of grain size vs. Ti showing all analyses where Fe was measured. Analyses with high Fe contents ($^{56}Fe/^{30}Si$ ratios
15   >0.007) are plotted in black. The coincidence of high Fe/Si ratios with high Ti suggests that the high Ti analyses were
contaminated by some non-quartz phase, possibly Fe-Ti oxides which are common in the Alpine Fault mylonites. All analyses
with $^{56}Fe/^{30}Si$ ratios >0.007 were removed from the dataset (see text for details).



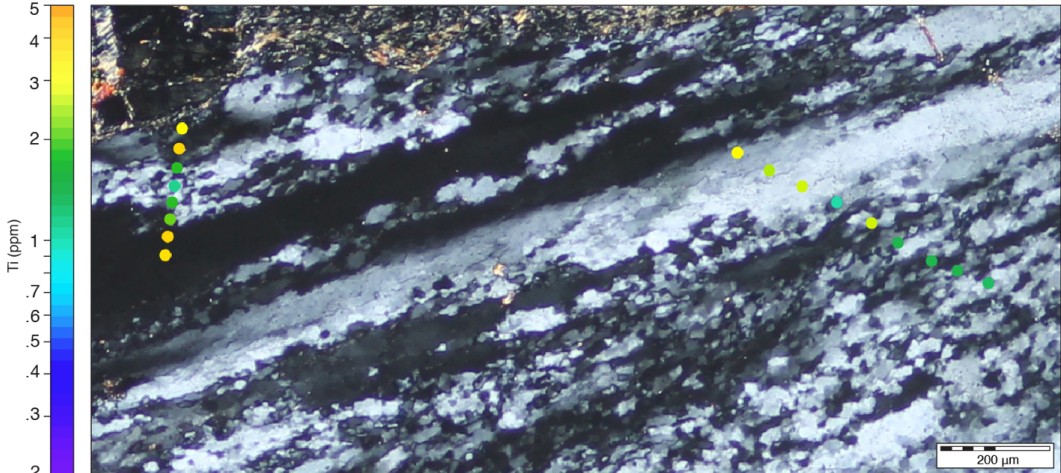

**Figure 3.** Cross polarized image showing dynamically recrystallized quartz in an ultramylonite sample (77912). Large ribbon grains, showing undulose extinction and subgrains, are interpreted as remnants, predating ultramylonite deformation. Recrystallized zones contain small, more equant grains with relatively uniform extinction. Typical of samples found within ~150 m of the Alpine Fault, lower Ti concentrations (green dots) are found in recrystallized zones than the ribbon grains (yellow and orange dots).





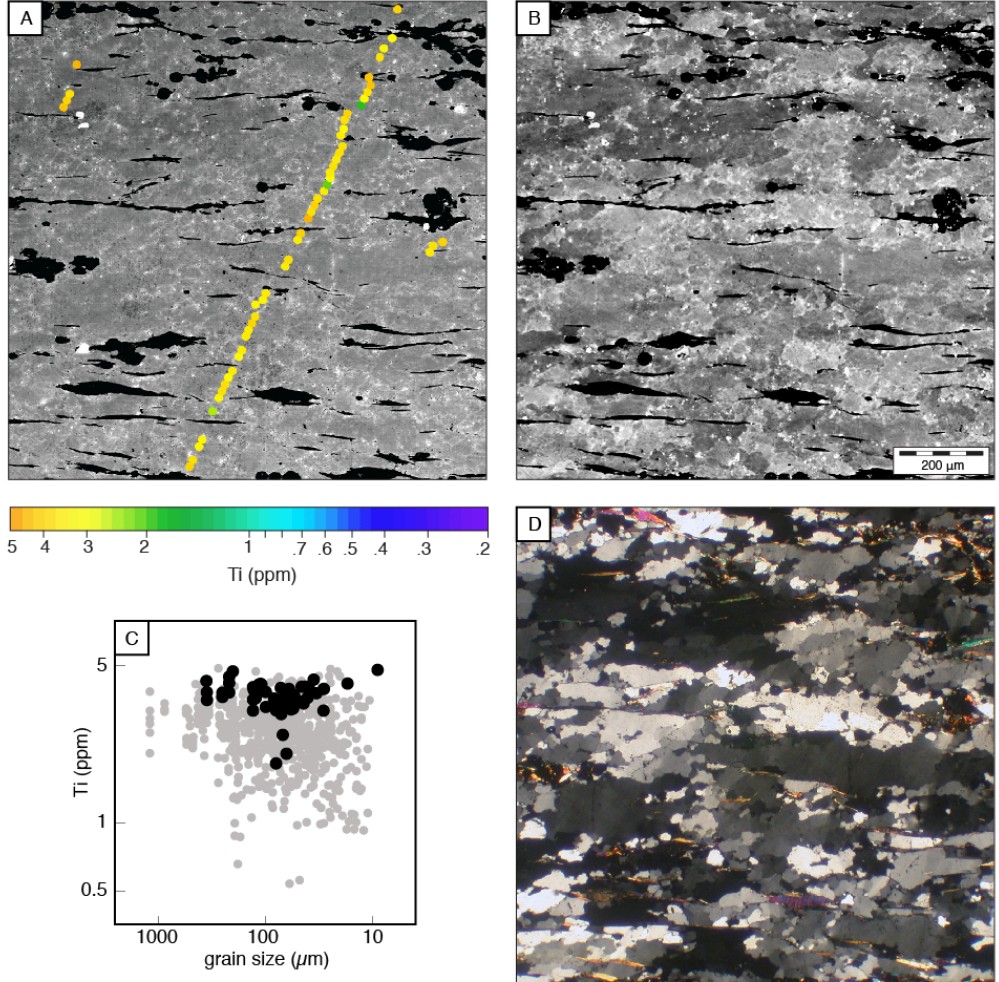

**Figure 4.** (A.) Panchromatic (200-960 nm) cathodoluminescence (CL) image of protomylonites metachert sample 77900 showing Ti concentrations at analysis points. (B) 470 nm wavelength cathodoluminescence (CL) image. (C) Ti-in-quartz data for this sample (black) against a backdrop of data from other Gaunt Creek samples—no variation in Ti content with grain size is observed. (D) Cross-polarized photomicrograph of the same area. As typical of many samples, Ti-in-quartz contents vary between ~2 and 3 ppm, with no apparent spatial pattern. The blue spectrum peak (~470 nm, 2.65 eV) correlates with quartz c-axis orientation such that grains with c-axes oriented perpendicular to the plane of the section are dark in panels (B) and (C). The Ti peak at ~415 nm (Wark and Spear, 2005) was not detected in this sample, probably due to the low concentrations of Ti in the sample.



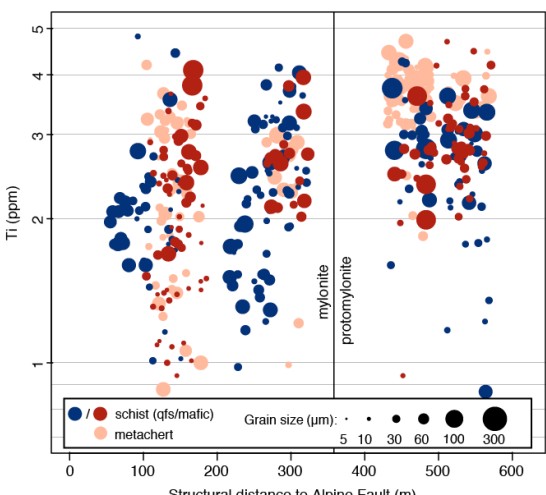

**Figure 5.** Ti concentrations vs. distance to the Alpine Fault, with analyses colored as a function of sample composition. With the exception of two protomylonite metachert samples with high Ti (at ~450 and 550 m), no systematic effect of composition is evident. The data are jittered along the x-axis to increase visibility.

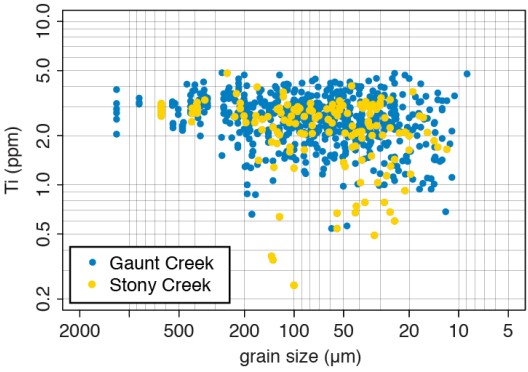

10  **Figure 6.** Plot of grain size vs. Ti concentration for all data. Grains >~250 μm have a relatively homogeneous Ti concentration ~2-4 ppm which we interpret to predate Alpine Fault deformation. Smaller grains (<200 μm) were at least partly formed by dynamic recrystallization during late stages of deformation, exhumation, and cooling. Despite very large strains (gamma = 12-120), less than half of the small recrystallized grains have Ti concentrations significantly lower than relict values of 2-4 ppm. We interpret that most deformation was associated with pressure-temperature conditions where Ti concentrations of 2-4 ppm were

15  stable, specifically that mylonite and protomylonite deformation occurred along a cooling path that followed a constant-Ti isopleth (see text for details). Stony creek analyses include data from Cross et al (2015).




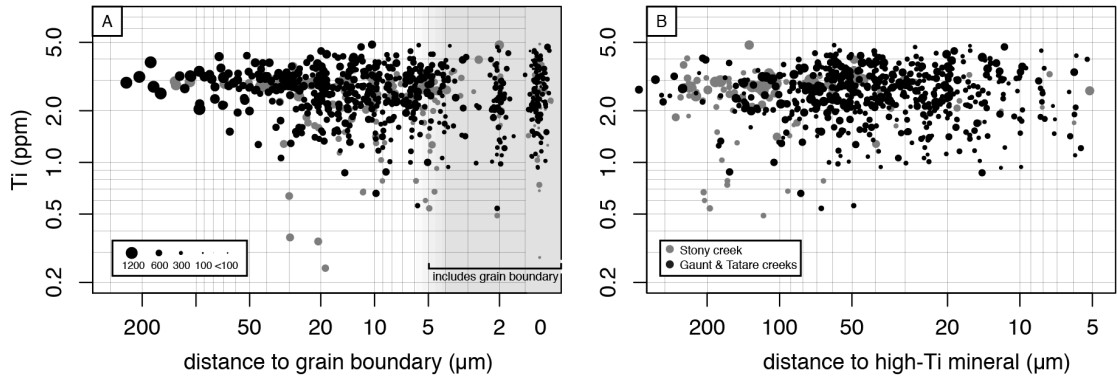

**Figure 7.** Ti concentrations in quartz vs. the distance (in thin section plane) from the analysis center to the nearest (A) grain
boundary (of any type) or (B) dark mineral that might act as a potential Ti source or sink such as biotite or Fe-Ti-oxide. The
10 increased presence of low Ti concentrations closer to grain boundaries (panel A) results from recrystallization and diffusion
during deformation at decreasing temperatures. The lack of a clear trend in (B) suggests that Ti-in-quartz concentrations were not
strongly affected by proximity to neighboring Ti-rich phases and that diffusion of Ti along grain boundaries was not a limiting
factor in Ti-in-quartz concentrations. The lack of low Ti quartz (<1 ppm) within 50 µm of high-Ti minerals (B) may relate to a
locally elevated Ti activity in the immediate vicinity of such minerals at low temperatures. Data from Cross et al. (2015) are
15 plotted in panel A but are not available for panel B. The data are jittered along the x axis to increase visibility.

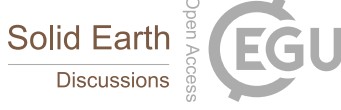



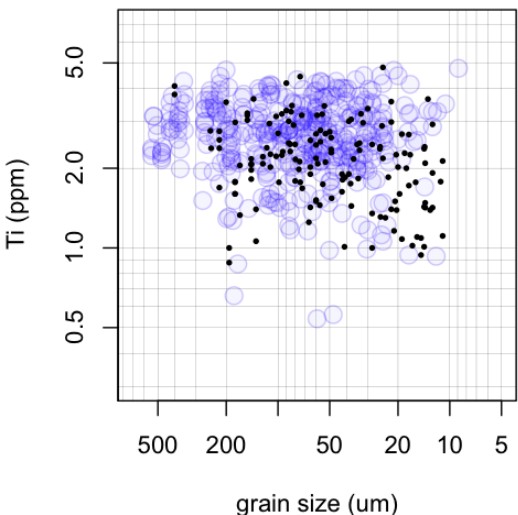

**Figure 8.** Grain size vs. Ti for Gaunt creek samples. Samples at structural distances <160 m from the fault are shown in black (mainly ultramylonites), while mylonites and protomylonites are plotted in transparent purple. The trend in the samples close to the fault suggests late recrystallization occurred at lower temperatures (with variable resetting). Away from the fault this trend is not observed—grain sizes tend to be larger, Ti tends to be greater, and, although it may occur in some samples, there is not a general trend of fine grains having lower Ti concentrations. In the ultramylonites, however, lower Ti concentrations (1-2 ppm) appear to have been stable during recrystallization of the finer grains. We interpret that deformation of the protomylonites and mylonites occurred during cooling that closely followed a Ti-isopleth (~2-4 ppm) in pressure-temperature space (e.g. Figure 11), but that ultramylonite deformation occurred at pressure-temperature conditions where Ti values of 1-2 ppm were stable. Only partial equilibration of Ti-in-quartz values was achieved during recrystallization of fine grains in the ultramylonite.





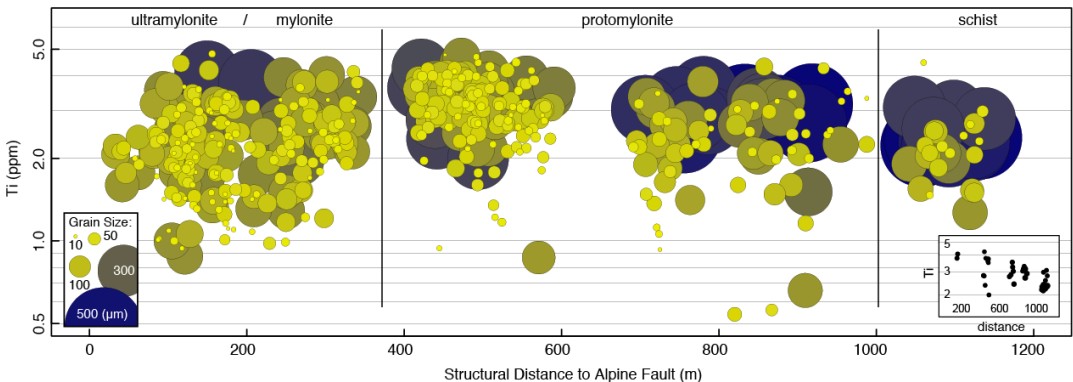

**Figure 9.** Ti concentrations vs. distance from the Alpine fault for Gaunt creek samples, with dot size indicating grain size. Small recrystallized grains (<~300 µm) near the fault are more likely to show decreased Ti concentrations than similar sized recrystallized grains away from the fault—this supports microstructural observations (e.g. Figure 3) that lower Ti concentrations 10 were stable during ultramylonite deformation but not during mylonite and protomylonite deformation. Inset shows Ti concentration in the coarsest grains (>300 µm) increases slightly towards the fault. Data are jittered along the x-axis.



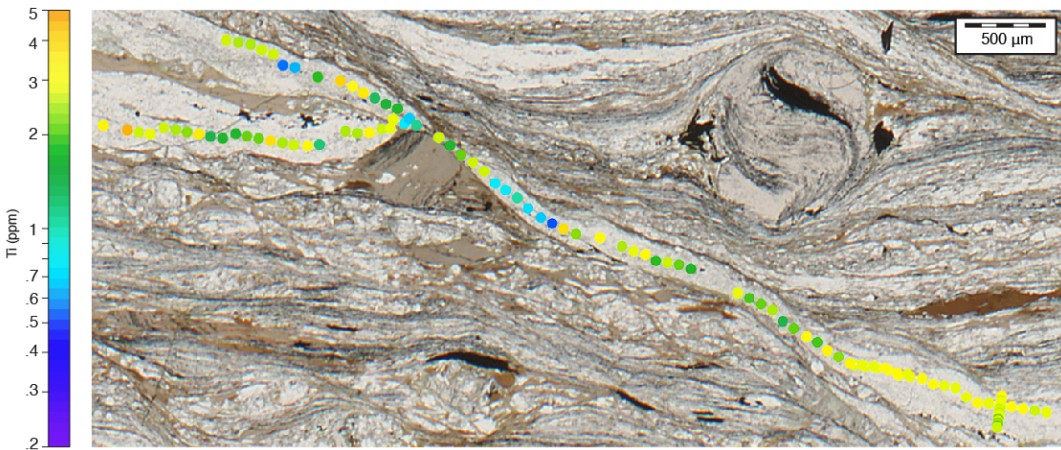

**Figure 10.** Image of an older foliation (near horizontal in this image) cut by a C' shear band in a sample from Stony creek. Ti
concentrations are plotted from both inside and outside the shear band. Most of the lowest Ti measurements are from within the
shear band, indicating some loss of Ti in quartz recrystallized during formation of the C' layer. The foliations preserved in the
garnet (upper right) and large biotite grain predate Alpine Fault deformation.



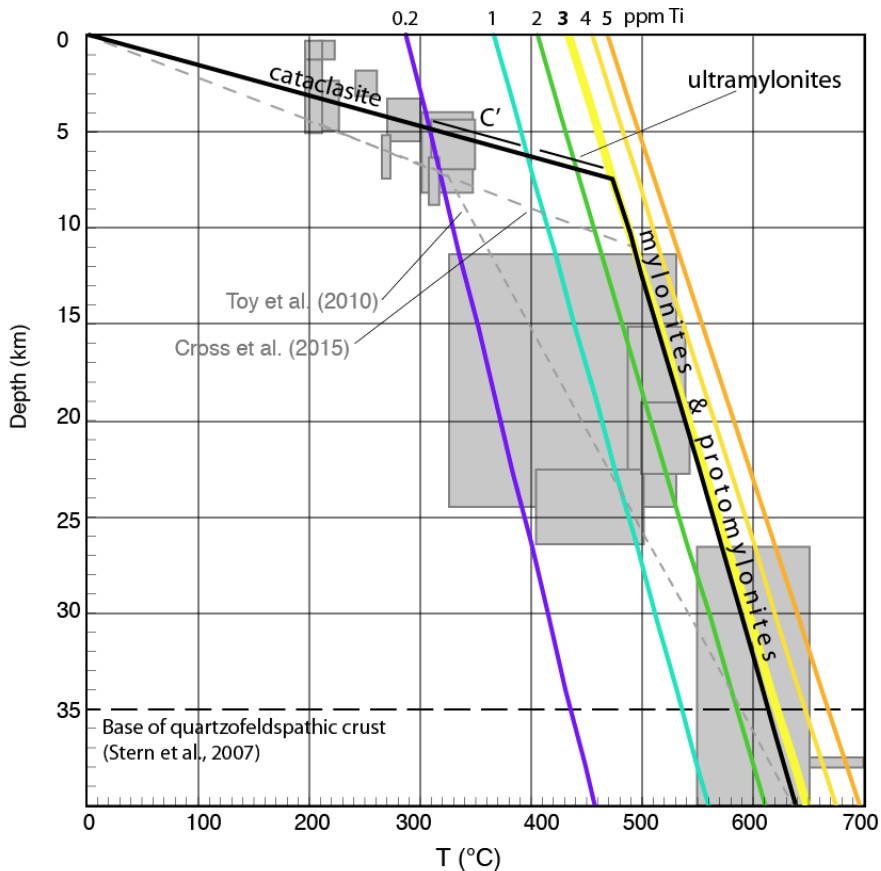

**Figure 11.** Temperature-depth diagram showing constraints on the Alpine Fault geothermal gradient from previous studies (grey boxes), proposed and previously proposed geothermal gradients (solid black and dashed lines, respectively), and experimentally determined constant-Ti contours (colored "isopleths") covering the range of observed Ti-in-quartz concentrations. A Ti activity of 0.1 (Thomas et al., 2010) and crustal density of 2.7 g/cm$^3$ are used to plot the isopleths. Ti activity is constrained by the fact that activities larger than 0.1 would shift the isopleths to the left and require quartz recrystallization at unrealistically low quartz recrystallization temperatures (<300 °C). The geotherm proposed here is based on new data indicating mylonite and protomylonite deformation occurred at temperatures from roughly 600 to 450 °C whilst Ti-in-quartz concentrations of ~3 ppm were stable. Previous temperature-pressure constraints are from Craw (1997), Holm et al. (1989), Green (1992), Grapes (1995), Cooper (1980), and Vry et al. (2004) as summarized by Toy et al. (2010).



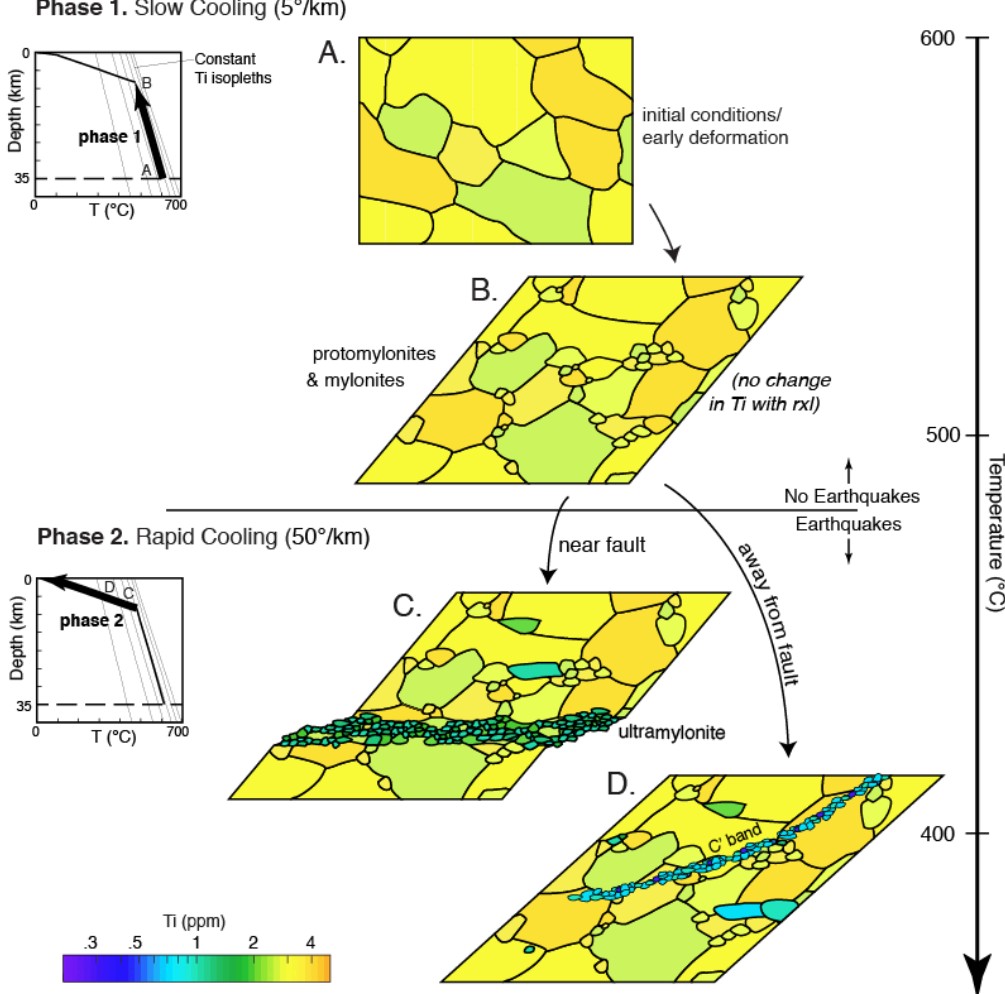

**Figure 12.** Simplified illustration of interpreted microstructural and Ti-in-quartz evolution. "Phase 1" deformation begins with
5    Ti-in-quartz concentrations at 2-4 ppm (A). Strain associated with protomylonite and mylonite formation (B) involves
recrystallization with no change in Ti-in-quartz concentrations due to exhumation along a constant-Ti isopleth (e.g. Figure 11).
During "phase 2," newly recrystallized quartz grains tend to have lower Ti concentrations as a result of cooling along a much
steeper segment of the geothermal gradient (Figure 11). Near the fault (C), ultramylonites form with Ti-in-quartz concentrations
of 1-2 ppm. The latest quartz recrystallization occurs away from the fault in protomylonites and mylonites in C' bands (D).
10   Sporadic recrystallization of some medium-sized grains with Ti-in-quartz concentrations below 2 ppm occurs in some rocks
during phase 2.