# Peer review of "Constraints on Alpine Fault (New Zealand) Mylonitization Temperatures and Geothermal Gradient from Ti-in-quartz Thermobarometry"

_Solid Earth, 2018_

## Referee Comment (RC1) · Anonymous Referee #1 · 3 Apr 2018

Review of "Constraints on Alpine Fault (New Zealand) Mylonitization Temperatures and Geothermal Gradient from Ti-in-quartz Thermobarometry" by Kidder, Toy, Prior, Little, and MacRae.

General Comments: This manuscript integrates microstructural and geochemical data on quartzites to interpret temperature conditions during deformation and mylonitisation of the Alpine Fault zone. The field locality provides an ideal setting to test the Ti-inquartz technique because there exist modern-day constraints on the rates, kinematics, and conditions of deformation from independent methods. The authors present an extensive dataset of Ti-in-quartz SIMS measurements reflecting a commendable

undertaking of skill and analytical investment. The SIMS results indicate very low Ti concentrations in all samples, regardless of many of the variables tested. The authors are faced with a challenging task in interpreting a PT history from these data and ultimately invoke the activity dependence on Ti solubility (assuming aTiO2=0.1) and argue that their samples followed a PT path that paralleled the trajectory of Ti-in-quartz isopleths (i.e., the samples experienced a protracted PT history that is not reflected in changes in Ti concentration because they coincidentally traveled along an isopleth, or line of constant concentration). While it may seem ad hoc to assume aTiO2=0.1 (even in samples with rutile present), there is increasing evidence in the literature to support activity values substantially less than unity, and applying this assumption to Kidder's dataset provides the best agreement with independent lines of evidence. Regardless of the details of their interpretation, the remarkable consistency of Ti-in-quartz measurements from different rock types with different deformation histories, proximity to fault zone, deformation mechanisms, etc., speaks to an efficient kinetic and equilibrium process that is homogenizing guartz Ti contents across a large volume of deforming rock. My primary critique is a lack of supporting information for the SIMS measurements. Not only do these measurements provide the basis of the manuscript, but they are just above analytical detection limits and thus require additional analytical details to ensure confidence. Overall the manuscript is well written and richly documented, and aside from a few areas of concern, I can offer only minor suggested revisions and pose several questions, answering of which may be outside the scope of this paper. Below I provide a summary of general observations from my reading followed by a list of detailed comments linked to line numbers in the text.

**Detailed Comments:**

Page 2 line 15 "places new constraints on the temperatures and pressures at which mylonitization occur at deep levels of the fault"

Comment: Consider re-wording. I suggest "at which mylonitisation occurred at deep levels of the Alpine Fault."
Page 3 line 35 "changes in biotite chemistry indicating a change in temperature during mylonitic deformation of roughly -340 °C"

Comment: Please add a reference for this statement

Page 4 line 5 "We used a regression line constrained though the origin to calculate Ti concentrations using National Institute of Standards (NIST) glasses 610 and 612 (434  $\pm$  15 and 44  $\pm$  5 ppm TiO2 respectively; Jochum et al., 2005)."

Comment: Note misspelling of "through". I recommend referencing the Jochum et al., 2011 GGR paper that publishes certified values for the NIST glasses instead of the 2005 paper that discusses the founding of the GeoReM database. Also, I would like to see more details regarding SIMS measurements added to this section. This is critically important given the very low values you measured from your samples. Please provide additional details on how you generated the SIMS calibration curve for your measurements – how many spots on the NIST glasses did you require in order to reduce precision on the calibration curve to acceptable levels? Did you make any measurements of the NIST glass as an unknown to evaluate the accuracy of your calibration curve? Why did you not also use the low concentration NIST glass SRM-614? Because your measurements are truly at the bottom of analytical detection limits, and your calibration curve is generated using two materials with considerably higher Ti concentration than your samples (434 ppm in NIST-610 and 44 ppm in NIST-612), the error on your calibration curve becomes magnified. It would be helpful if you plotted your calibration curve in the Supplementary Materials.

Page 4 line 10 "No blank correction was made since this value is far below Ti concentrations measured in the study."

Comment: Not entirely true. The upper confidence bounds of the Herkimer measurement ( $\sim$ 105 ppb) are  $\sim$ 40% of your lowest measurement results (240 ppb). Even if you don't apply the blank correction because it makes concentrations even lower than they already are, I would consider omitting the above sentence.
Page 4 line 16 "We interpret that all, or nearly all, of the analyses with Ti >5 ppm were contaminated, possibly by trace amounts of Fe-Ti oxides"

Comment: This data filtering step needs to be better documented. What percentage of your measurements were omitted based on this 5 ppm cutoff? Do you have evidence from your EDS particle analysis that supports Fe-Ti oxides in samples with elevated Fe/Si ratios? With such high mass resolving power why do you think there would be interferences of Fe mass 56 on the Ti mass 47 and 49 measurements? Do you see this same contamination effect on both Ti masses?

Page 4 line 35 Comment: I would like to see more details of the results of the automated EDS particle analysis "Feature". As far as I can tell the only mention of these results is y/n for rutile or chlorite in Table 1. How much did rutile content vary between samples? Was there any relationship between rutile abundance and sample Ti concentration? It would be helpful to plot this in similar fashion to grain size vs. Ti conc. plots. This would provide evidence for your activity assumption, and could also shed light on discussion involving local activity buffering (absence of low-Ti measurements in proximity to rutile grains). (I later saw Figure 7: Which minerals are included as "dark minerals"? Biotite, ilmenite, rutile?)

Page 6 line 18 "A paucity of Ti values

mylonitization temperature of ~550°C (Cross et al., 2015; Toy et al., 2010) is 6 b.y.,"

Comment: Technically you should use the grain radius (since diffusion would progress from the grain centre to the margin) as the effective diffusion distance instead of the diameter, yielding 3 b.y. (your point remains).

Page 9 line 36 "The question of Ti activity is further hindered by competing calibrations of TitaniQ (Huang and Audétat, 2012; Thomas et al., 2010; Thomas et al., 2015)."

Comment: Consider re-wording. Important to note here for the readers that the Huang and Audetat calibration has no activity dependence.

Page 10 line 19 "An absence of Ti values less than  ${\sim}1$  ppm at distances

---

## Short Comment (SC1) · 17 Apr 2018

D. Grujic

dgrujic@dal.ca

Received and published: 17 April 2018

It was a pleasure to read your manuscript. It is very nicely and concisely written. The research is perfectly conceived and meticulously executed. Consequently, you left me very little space for comments. My principal comment is a question: Is there a reason you did not try to determine the activity of TiO2 using the approach by Ashley and Law (2015)? Second comment is a suggestion for Conclusion (P 11, line 22): Could you instead of fluid circulation within the shear zone consider that the high temperatures exist(ed) at 8 km and bellow, higher than in a typical continental setting although with the low geothermal gradient, while the upper 8 km of the crust is a thermal boundary layer

caused by the uplift of relatively hotter rocks by the reverse component of the Alpine Fault? Overthrusting can in general induce a near-surface thermal boundary layer, the gradient in it being the function of thermal properties of rocks and rock exhumation rate.

Minor comments: 1. Please replace "Ti activity" with "activity of TiO2 (aTiO2)" 2. Please label consistently the temperature with symbol °C and put a space between the numeric value and the symbol. 3. Please use consistently the appropriate term "geothermal gradient" or "geotherm". P3, line 1: replace "schist foliation" with schistosity P3, line 5: delete either contractional or shortening P 3, line 10: please put the Greek letter gamma in front of the value for the simple shear strain. P 3, line 13: All that indicates a general strain. Are there vorticity analyses for the mylonites from the Alpine fault? In addition, you first write flattening and at the end of the sentence stretch. What was the geometry of the strain: flattening, plane strain or constriction? P 3, line 35: Consider removing minus sign in front of 340 °C and changing the wording accordingly. P 3, line 37: I prefer not to use adjective "extensional" with "shear bands". C' are always dipping in direction of displacement, both in thrust- and normal faultgeometry shear zones (i.e., contractional and extensional settings). P 5, line 30: no need for quotation marks with CL P 5, line 33: why "apparent" rims? P 5, line 35: are garnets porphyroclasts or rather porphyroblasts? P 7, line 10: consider rewording to "wide temperature range from 360 °C to 500 °C" P 7, line 34: double check if 6 billion years is correct value and correct abbreviation. P 9, line 13: could such a difference in geothermal gradient, at relatively small distance, be steady over a geological time scale relevant to this study, unless the cause for it was active? P 10, line 22: in the previous sentence activity of TiO2 of 0.1 was indicated, and with wording in this line it reads as if it were a high value. P 11, line 22: is there independent evidence for fluid (meteoric or metamorphic) during dynamic recrystallisation? Figure 12: consider flipping the vertical axis for the main part of the figure to be consistent with the insets (temperature decreases upward). In addition, is this rapid cooling or high geothermal gradient?

СЗ

---

## Short Comment (SC2) · 17 Apr 2018

I forgot to mention that while preparing the report I took in account the technical comments by reviewer #1, which, to my understanding, are all pertinent.

---

## Referee Comment (RC2) · Anonymous Referee #2 · 2 May 2018

The comment was uploaded in the form of a supplement:
https://www.solid-earth-discuss.net/se-2018-12/se-2018-12-RC2-supplement.pdf
* * *

---

## Author Comment (AC1) · 29 Jun 2018

The authors present an extensive dataset of Ti-in-quartz SIMS measurements reflecting a commendable undertaking of skill and analytical investment.

Thank you

My primary critique is a lack of supporting information for the SIMS measurements. Not only do these measurements provide the basis of the manuscript, but they are just above analytical detection limits and thus require additional analytical details to ensure confidence.

We now provide more information on the routine (see below). We also note that it was not made sufficiently clear that the SIMS measurement procedure followed that of Kidder et al., 2013 (Solid Earth), which provided significantly more documentation and verification of the analytical procedure than given here. In the revised version we explicitly state this, and provide additional information such as the calibration curve and analyses of the standards (in a supplement).

The detection limit of the SIMS instrument, incidently, is far below even the lowest values we measured. For example, in 2013, we determined a Ti-in-quartz concentration of 4.5 +/-2 ppb for the Herkimer quartz standard using nearly the same analytical setup as in the current study. The lowest Ti quartz analyzed as part of the current study contained about two orders of magnitude more Ti.

Detailed Comments:

Page 2 line 15 "places new constraints on the temperatures and pressures at which mylonitization occur at deep levels of the fault"

Comment: Consider re-wording. I suggest "at which mylonitisation occurred at deep levels of the Alpine Fault."

Ok, changed the wording as suggested

Page 3 line 35 "changes in biotite chemistry indicating a change in temperature during mylonitic deformation of roughly −340 $^{\circ}$C"

Comment: Please add a reference for this statement

We moved the reference to Toy et al. (2008, 2010) to the end of the sentence to clarify where this number came from.

Page 4 line 5 "We used a regression line constrained though the origin to calculate Ti concentrations using National Institute of Standards (NIST) glasses 610 and 612 (434 ± 15 and 44 ± 5 ppm TiO2 respectively; Jochum et al., 2005)."

Comment: Note misspelling of "through". I recommend referencing the Jochum et al., 2011 GGR paper that publishes certified values for the NIST glasses instead of the 2005 paper that discusses the founding of the GeoReM database. Also, I would like to see more details regarding SIMS

measurements added to this section. This is critically important given the very low values you measured from your samples. Please provide additional details on how you generated the SIMS calibration curve for your measurements – how many spots on the NIST glasses did you require in order to reduce precision on the calibration curve to acceptable levels? Did you make any measurements of the NIST glass as an unknown to evaluate the accuracy of your calibration curve? Why did you not also use the low concentration NIST glass SRM- 614? Because your measurements are truly at the bottom of analytical detection limits, and your calibration curve is generated using two materials with considerably higher Ti concentration than your samples (434 ppm in NIST-610 and 44 ppm in NIST-612), the error on your calibration curve becomes magnified. It would be helpful if you plotted your calibration curve in the Supplementary Materials.

We corrected the misspelling and changed the reference as suggested. Twelve analyses of NIST glass were acquired and used for the calibration. This information is now added to the methods section. We now provide the calibration diagram and data in supplementary materials. We did not use the lower Ti NIST standard mainly because it was not available in the Caltech SIMS lab. Because SIMS measurements of quartz show linear increases in Ti+/Si+ ion ratios with increasing Ti content (e.g. Behr et al., 2010) we would not expect a significant change in the calibration from using a standard with lower Ti content.

As mentioned above, the detection limit (and background) of Ti on the SIMS instrument used is much lower than the ppm levels, e.g. as demonstrated by Kidder et al., 2013 who measured the Ti concentration of Herkimer quartz to 4.5 +/-2 ppb. In the current study, we deliberately shortened acquisition time in order to collect more data at the expense of somewhat noisier results. Using the shortened acquisition time, most our measurements (~2-3 ppm) are still more than an order of magnitude above the levels measured in the Herkimer quartz "blank" (~0.08 ppm, as mentioned in the text).

Finally, the same instrument and protocol were used in the Kidder et al. (2013) study. In that study, we tested the calibration curve against quartz standards obtained from Jay Thomas and found results within error of the independently-obtained values. The difference between the current and 2013 calibration is insignificant in terms of the final temperatures that are calculated (about 1 degree difference).

We did not use any NIST measurements as an unknown, but inspection of the calibration curve in the supplement shows that the standards measurements were quite uniform.

Page 4 line 10 "No blank correction was made since this value is far below Ti concentrations measured in the study."

Comment: Not entirely true. The upper confidence bounds of the Herkimer measurement (~105 ppb) are ~40% of your lowest measurement results (240 ppb). Even if you don't apply the blank correction because it makes concentrations even lower than they already are, I would consider omitting the above sentence.

We changed it to simply say "No blank correction was made." (removed "since this value is far below Ti concentrations measured in the study.")

Page 4 line 16 "We interpret that all, or nearly all, of the analyses with Ti >5 ppm were contaminated, possibly by trace amounts of Fe-Ti oxides"

Comment: This data filtering step needs to be better documented. What percentage of your measurements were omitted based on this 5 ppm cutoff? Do you have evidence from your EDS particle analysis that supports Fe-Ti oxides in samples with elevated Fe/Si ratios? With such high mass resolving power why do you think there would be interferences of Fe mass 56 on the Ti mass 47 and 49 measurements? Do you see this same contamination effect on both Ti masses?

52 data points, or ~7% of the measurements were removed (this has been added to the text). And, yes, all of the samples contain Fe-Ti oxide. We do not mean to suggest that there was interference (almost certainly there wasn't)—simply, if a trace amount of Fe-Ti oxide were analyzed along with quartz, the analysis would show both elevated Fe and Ti levels. We interpret that the elevated Fe counts associated with elevated Ti values indicate this scenario, not actual high Ti sourced from the crystal lattice of quartz. (Possibly one might correct for this by subtracting out equivalent contents of the Fe and Ti to isolate the quartz concentration, however we also observe in backscatter images that Fe-Ti oxides are sometimes intermingled with variable amounts of a pure Ti phase.)

We've added to the text that "small grains of Fe-Ti oxide are common in all of the samples"

Page 4 line 35 Comment: I would like to see more details of the results of the automated EDS particle analysis "Feature". As far as I can tell the only mention of these results is y/n for rutile or chlorite in Table 1. How much did rutile content vary between samples? Was there any relationship between rutile abundance and sample Ti concentration? It would be helpful to plot this in similar fashion to grain size vs. Ti conc. plots. This would provide evidence for your activity assumption, and could also shed light on discussion involving local activity buffering (absence of low-Ti measurements in proximity to rutile grains). (I later saw Figure 7: Which minerals are included as "dark minerals"? Biotite, ilmenite, rutile?)

We added this text (section 6.1) "In samples containing rutile, it's abundance relative to other oxides varied from about 2% to 97%, but these values are not correlated with average Ti concentrations of the samples." We made the plot but it doesn't show a trend so don't think it should be added as a figure. We also now list (section 6.1) the most common "dark" minerals observed: "biotite, hornblende, ilmenite, magnetite or rutile"

Page 6 line 18 "A paucity of Ti values <1 ppm Ti at distances <5 μm may be a function of minor contamination of analyses that include grain boundaries (figure 7a)"

Comment: This is counter to my intuition. I would expect partial analysis of grain boundaries to, if anything, result in higher Ti measurements (instead of lower) because of the possibility of intersecting a Ti-bearing phase along the grain boundary. It makes less sense to assume that a low measurement reflects a partial grain boundary measurement because if Ti counts were low

because you were in the boundary then the Si counts would be correspondingly low as well.

We had the same understanding, however the wording wasn't clear. We infer that low Ti quartz was analyzed along grain boundaries, but that it was contaminated (brought to higher levels) such that we don't actually get any low-Ti measurements of grain boundaries. We have changed the wording to clarify our meaning here.

Page 7 line 32 "The characteristic diffusion timescale (e.g. Spear, 1995) for 250 μm grains using the experimental data of Cherniak et al. (2007) at a peak Alpine Fault mylonitization temperature of ~550°C (Cross et al., 2015; Toy et al., 2010) is 6 b.y.,"

Comment: Technically you should use the grain radius (since diffusion would progress from the grain centre to the margin) as the effective diffusion distance instead of the diameter, yielding 3 b.y. (your point remains).

Good point. And actually for a distance of 125 microns the time scale is 1.5 b.y. We have modified the text accordingly

Page 9 line 36 "The question of Ti activity is further hindered by competing calibrations of TitaniQ (Huang and Audétat, 2012; Thomas et al., 2010; Thomas et al., 2015)."

Comment: Consider re-wording. Important to note here for the readers that the Huang and Audetat calibration has no activity dependence.

The formulation presented by Huang and Audetat does not contain an activity variable, however we believe they intended that a simple correction for activity be made and implemented in their formulation (using the product of Ti concentration and Ti activity in place of Ti concentration). This was previously done in several publications already, e.g. Cross et al., 2015; Audetat, 2013; and implicit in Huang and Audetat (2012) since they tested their calibration using quartz with non-unity Ti activity. It seems clear to us that Huang and Audetat did not intend that their calibration only be used in cases where Ti activity was 1.0, and assumed that users would make the correction we have made.

In any case, the point made in our paragraph is that not only do we often not know Ti activity, but even in cases of quartz crystallized at a well-constrained temperature, uncertainty is introduced by the existence of two TitaniQ calibrations. We have modified the wording slightly to clarify this.

Page 10 line 19 "An absence of Ti values less than ~1 ppm at distances <50 μm from phases containing stoichiometric quantities of Ti (Figure 7B) may indicate such variation"

Comment: Confusing as written. Not sure if you are referring to the previous sentence (that temperature might affect Ti activity) or the sentence before that (that activity might vary spatially within a sample). You don't have any evidence to suggest that activity varies with temperature, and indeed this would complicate your interpretation that exhumation followed the ~3 ppm isopleth with activity=0.1.

We were suggesting both (variation locally but only at low temperature). We've clarified this and heavily edited the paragraph to present the idea more clearly. It now reads "While it is convenient to assign a single value of $a_{TiO2}$ to a suite of rocks such as our sample set—and the overall consistency of Ti-in-quartz concentrations in different lithologies generally supports this (Figure 5)—sample- or micro-scale variations in $a_{TiO2}$ due to local differences in bulk or fluid composition may exist. Additionally, activity of $TiO_2$ could also vary with temperature (Ashley and Law, 2015). An absence of Ti values less than ~1 ppm at distances <50 μm from phases containing stoichiometric quantities of Ti (Figure 7B) hints at both types of $a_{TiO2}$ variation. The absence of Ti values <1 ppm could be explained if $a_{TiO2}$ was higher than 0.1 in the vicinity of high-Ti minerals during the latest phase of quartz recrystallization. A change in $a_{TiO2}$ to a value of 0.5 in the immediate vicinity (<50 μm) of Ti-bearing phases at temperatures of 300 °C would stabilize Ti concentrations at 1 ppm and explain the local absence of lower Ti measurements. The extent of recrystallization at these conditions was relatively minor, as suggested by the low frequency of such analyses (Figure 7) and their location only within 30 μm of grain boundaries (Figure 7a)."

Page 10 line 26 Comment: Include units for strain rate

Done

Page 11 line 10 "whereas we hypothesize the partial preservation of Ti concentrations and individual larger grains from the entire time period including and predating Alpine Fault deformation."

Comment: Consider re-wording

Changed to "whereas we hypothesize that larger grains were preserved from before and throughout Alpine Fault deformation and that these grains record a history of equilibrium Ti concentrations that happen to have varied little over time."

Comment: I recommend ordering references chronologically instead of alphabetically, this allows the reader to see the progression of work on a particular topic.

We agree and changed the order accordingly.

---

## Author Comment (AC2) · 29 Jun 2018

>It was a pleasure to read your manuscript. It is very nicely and concisely written. The research is perfectly conceived and meticulously executed.

Thank you

>Consequently, you left me very little space for comments. My principal comment is a question: Is there a reason you did not try to determine the activity of TiO2 using the approach by Ashley and Law (2015)?

The approach of Ashley and Law (2015) is useful for cases where rutile is absent. When rutile is present, as is the case in most of our samples, their approach would simply give TiO2 activity = 1. For rutile-absent samples under our PT conditions, the Ashley and Law (2015) approach also suggests high Ti activities close to 1, which seems supported by our observation of no noticeable difference in Ti concentrations between samples with and without rutile. As discussed in section 7.2, a Ti activity value of 1.0 is much higher than seem possible given independent constraints on the lower limit of quartz recrystallization temperature. Alternatively, using the Huang and Audetat (2012) TitaniQ calibration would provide temperatures consistent with higher Ti activity values (0.5-0.8, as mentioned in the text), however Ashley and Law (2015) provide a number of arguments against using that calibration. As mentioned by the first reviewer, "there is increasing evidence in the literature to support activity values substantially less than unity." More work is needed on this issue of TiO2 activity, but it is beyond the scope of our paper.

Second comment is a suggestion for Conclusion (P 11, line 22): Could you instead of fluid circulation within the shear zone consider that the high temperatures exist(ed) at 8 km and bellow, higher than in a typical continental setting although with the low geothermal gradient, while the upper 8 km of the crust is a thermal boundary layer caused by the uplift of relatively hotter rocks by the reverse component of the Alpine Fault? Overthrusting can in general induce a near-surface thermal boundary layer, the gradient in it being the function of thermal properties of rocks and rock exhumation rate.

Yes, thank you for the suggesting this possibility. Looking into this further, we agree that this is a likely origin of the "kink" we describe. We now point out that the pattern in the geothermal gradient we describe is similar to that predicted by 2D thermal models of rapid exhumation that do not involve fluid circulation (adding the Koons, 1987 numerical modeling reference). At the conclusion of the paragraph we now leave the matter more ambiguous, stating "The position of the kink in this study corresponds roughly to the maximum depth of earthquakes near the study area (Boese et al., 2012; Leitner. et al., 2001), suggesting that the onset of rapid cooling reduces temperatures to conditions where brittle processes dominate, and/or that fracturing associated with seismicity accentuates cooling by the infiltration of meteoric fluids to deep levels (Menzies et al., 2014)." The Menzies reference was added to indicate that there is evidence of meteoric fluids reaching these depths.

Minor comments: 1. Please replace "Ti activity" with "activity of TiO2 (aTiO2)"

Done

2. Please label consistently the temperature with symbol $^\circ$C and put a space between the numeric value and the symbol.

Done

3. Please use consistently the appropriate term "geothermal gradient" or "geotherm".

Done

P3, line 1: replace "schist foliation" with schistosity

Done

P3, line 5: delete either contractional or shortening

Done

P 3, line 10: please put the Greek letter gamma in front of the value for the simple shear strain.

Done

P 3, line 13: All that indicates a general strain. Are there vorticity analyses for the mylonites from the Alpine fault? In addition, you first write flattening and at the end of the sentence stretch. What was the geometry of the strain: flattening, plane strain or constriction?

We've clarified the language and added the requested information. It now reads "Characteristics such as boudinaged quartzose layers parallel to the shear zone, and reorientation of inherited, pre-Neogene lineations indicate that mylonitization involved ductile thinning by a factor >3 (Gillam et al., 2013; Norris and Cooper, 2003; Toy et al., 2013). Kinematic vorticity number ($W_k$) is estimated to be 0.7-0.85 in the mylonite zone (Little et al., 2016) or higher (Toy et al., 2013). Strain in the mylonite likely had a flattening geometry, but with $S_1$ exceeding $S_2$ by ~30 times (Toy et al., 2013)."

P 3, line 35: Consider removing minus sign in front of 340 $^\circ$C and changing the wording accordingly.

Good idea. We changed it to a "decrease in temperature…" and removed the minus sign

P 3, line 37: I prefer not to use adjective "extensional" with "shear bands". C' are always dipping in direction of displacement, both in thrust- and normal fault- geometry shear zones (i.e., contractional and extensional settings).

Sounds good. We removed "extensional."

P 5, line 30: no need for quotation marks with CL

Removed

P 5, line 33: why "apparent" rims?

Removed "apparent"

P 6, line 35: are garnets porphyroclasts or rather porphyroblasts?

Porphyroclasts (no change)

P 7, line 10: consider rewording to "wide temperature range from 360 °C to 500 °C"

We made this change.

P 7, line 34: double check if 6 billion years is correct value and correct abbreviation.

We adjusted this value in response to the first reviewer (see above).

P 9, line 13: could such a difference in geothermal gradient, at relatively small distance, be steady over a geological time scale relevant to this study, unless the cause for it was active?

We are not sure, but think variations in geotherm at this length scale are plausible. Variable dip angle of the Alpine Fault at a short wavelength has been proposed as an explanation for large variation in exhumation rate over short distances (e.g. Little, 2005 GSA Bulletin), and such changes could affect geothermal gradients. Similarly, upper crustal geothermal gradients have been demonstrated to vary substantially at short wavelengths in the recent drill holes (Sutherland et al., 2017).

P 10, line 22: in the previous sentence activity of TiO2 of 0.1 was indicated, and with wording in this line it reads as if it were a high value.

We have changed this paragraph significantly (see above response to other reviewer) and this sentence is no longer present.

P 11, line 22: is there independent evidence for fluid (meteoric or metamorphic) during dynamic recrystallisation?

Yes, Menzies et al. (2014). We've now added this reference.

Figure 12: consider flipping the vertical axis for the main part of the figure to be consistent with the insets (temperature decreases upward). In addition, is this rapid cooling or high geothermal gradient?

Yes, good idea, we have implemented this in Figure 12.

Assuming constant uplift rate, this is both rapid cooling and high geothermal gradient. We've changed the figure text to indicate this, e.g. "60°/km geotherm, rapid cooling"

---

## Author Comment (AC3) · 29 Jun 2018

The uploaded document duplicates the original comments

---

## Editor Decision (ED1)

**Editorial notes (need to be corrected or resolved):**

Page 1 - line 18 Please capitalize "moho"

starting from Page 2 - line 33 In running text, please consistently use "Fig." - not "Figure".

Page 3 - line 30 one space too many after end of sentence

Page 4 - line 30 one space too many after end of sentence

Page 4 - line 35ff Could you be a bit more specific as to how you define and determine "grain size" ?

Page 5 - line 18 It would be important to know what is the grain size here, in a case where you have elongated grains. In Fig.3 and Fig.4c, you may want to specify "long diameter" or "equivalent diameter" or whatever it is...

Page 5 - line 33 Much as I look at Fig.4a and 4b. The rims look bright to me (as they should, being depleted) - not dark. Also, please refer to both Fig.4a and 4b.

Page 7 - line 37 Please use Ga instead of byr.

Figure 4

Why did you use uppercase A,B,C,D while references to image are lowercase? I am not sure if Solid Earth has a policy about using upper- or lowercase.

**Page 21 - line 6 - Figure 4 - caption**

As Fig.4d show a cross-polarized photomicrograph (...), the observation that "grains with c-axes oriented perpendicular to the plane of the section are dark in panels (B) and (D)" cannot strictly be upheld. Whether or not a grain appears dark on a cross-polarized photomicrograph not only depends on the inclination of the quartz c-axis w/ r to the image plane, but also on the orientation w/r to the horizontal and vertical direction within the image plane. In fact you also find dark grains in Fig. 4d that are not dark in Fig. 4d. If you had used circular polarization, the correlation between Fig. 4b and 4d would be perfect.

Please modify or delete this remark from the caption.

Page 22 - line 3 Figure 5 "The data are jittered along the x axis to increase visibility." As you are plotting Ti concentrations in quartz vs. the distance, this sentence makes no sense here. Please delete.

Page 22 - line 11 Figure 6.

No, you are not plotting "grain size vs. Ti concentration" but the reverse. One always plots Y against X (where X is the independent variable and usually the horizontal), Please correct.

Page 23 Figure 7 (cf. Figure 4) Why did you use uppercase A,B while references to image are lowercase? I am not sure if Solid Earth has a policy about using upper- or lowercase.

Page 23 - line 13 Figure 7 (cf. Figure 5) "The data are jittered along the x axis to increase visibility." Again, as you are ostensibly plotting Ti concentrations in quartz vs. the distance, this sentence makes no sense here. Please delete.

Page 24 - line 11 Figure 8 Again, you are not plotting "grain size vs. Ti concentration" but the reverse. (see Fig. 6) Please correct.

Page 25 - line 11 Figure 9 "Data are jittered along the x-axis." Same as in Figure caption 7, this sentence makes no sense here. Please delete. Instead, please comment on the inset.

Page 26 - line 4 Figure 10 "Image" ? Please specify.

**Comments (respond at your own discretion):**

Page 1 line 31 you say "...where the examination of plate boundary phenomena ... can be informed by observation ...." getting close to Yogi Berra's famous quote: "You can observe a lot by just watching." :-) Page 2 - line 30

you say: "classic sequence of fault rocks: protomylonites, mylonites, ultramylonites, and finally cataclasites" - cataclasites are "classical" if the fault development is following a cooling path or later-stage brittle overprint....

**Page 3 - line 3**

your write: "clockwise rotation (when viewed from the SW) and dextral shearing..." - I am not familiar with the details of the Alpine fault deformational history... So I wonder: are "clockwise rotation" and "dextral shearing" two different events or both the same? Specifying the viewing direction ("when viewed from the SW") indicates to me that the "rotation" must be on a vertical or very steep plane, and presumably applies to the \*dextral shearing" too. If so, and if both "clockwise rotation" and "dextral shearing" are the same you might consider calling this " top to the NE shearing" and save yourself the viewing direction...

Page 3 - line 10 What is the basis for the strain determinations? Any references ?

Page 3 - line 38 (Page 4 - line 1 why "common in either" - why not "common in both" or simply "common in" ?

Page 23 - line 10-13

Figure 7

Very long caption. "The lack of a clear trend in (B) suggests that..." etc. until "... not available for panel B."

You may consider inserting (and possibly shortening) this part of the caption into the running text on page 6 - line 19.

Page 24 - line 5-11 Figure 8 Very long caption. "Away from the fault this trend..." etc. until "... of fine grains in the ultramylonite." I think it would be better to move this descriptive part of the caption to the running text on page 6 - line 35.

Page 27 - line 5-10 Figure 11 Very long caption. "Ti activity is constrained by the fact that..." etc. until "... as summarized by Toy et al. (2010)." Consider moving this part of the caption to the running text on page 8 - line 9.

Figure 12

In a time where it has become fashionable to attribute every brittle microstructure to an earth quake, I would welcome if you could label the horizontal line in Figure 12 as "limit of seismicity" (as in text) and remove "Earthquakes" and "No Earthquakes". -Else, the next thing you know is somebody citing your paper as a demonstration of earthquakes as producing ultramylonites with low Ti-in-quartz values.

---

## Author Response (AR2)

Dear Renée,

Thank you for your assistance with the paper and giving it a close read yourself. I've made the changes recommended in the editorial notes and most of those in the comments. Please see details below.

Thank you,

Steve Kidder

Editorial notes (need to be corrected or resolved):

Page 1 - line 18 - Please capitalize "moho"

Changed

starting from Page 2 - line 33 - In running text, please consistently use "Fig." - not "Figure".

OK, Hopefully I did this correctly. I changed both instances in parentheses and also those in the main text for single and plural to "Fig." For example, "This is demonstrated in Fig. 3, but also seen elsewhere (e.g. Fig. 5, 6, 7)"

Page 3 - line 30 - one space too many after end of sentence

Done

Page 4 - line 30 - one space too many after end of sentence

Done

Page 4 - line 35 - Could you be a bit more specific as to how you define and determine "grain size" ?

Yes. We added the following sentence to the last paragraph of the methods section: "Grains were distinguished from subgrains based on the sharpness of grain boundaries under cross-polarized light."

Page 5 - line 18 - It would be important to know what is the grain size here, in a case where you have elongated grains. In Fig.3 and Fig.4c, you may want to specify "long diameter" or "equivalent diameter" or whatever it is...

We added the following sentence to the last paragraph of the methods section: "Long and short axes of elongate grains were measured, and reported sizes for these grains are diameters of circles with the same area as an ellipse having the measured axes lengths."

Page 5 - line 33 - Much as I look at Fig.4a and 4b. The rims look bright to me (as they should, being depleted) - not dark. Also, please refer to both Fig.4a and 4b.

We agree, the rims in figure 4 are bright. But the image in figure 4a and 4b are not "typical" CL images, they are the hyperspectral CL images (which we wanted to show for other reasons...).

We've added information explaining this explicitly now: "Typical CL images of the Alpine Fault mylonites and protomylonites show a homogeneous illumination (e.g. Fig. 4a), generally with a thin dark rim around the edges of grains (note that this relationship is reversed in the hyperspectral CL images in Fig. 4a and 4b that show brighter rims).

Page 7 - line 37 - Please use Ga instead of byr.

Fixed

Page 21 - Figure 4 - Why did you use uppercase A,B,C,D while references to image are lowercase? I am not sure if Solid Earth has a policy about using upper- or lowercase.

We're confused here. We did refer to the images in the figure caption using uppercase, i.e. the same as is shown in the upper corners of the figures. For example the caption says "...in panels (B) and (D)..."

Page 21 - line 6 - Figure 4 – caption - As Fig.4d show a cross-polarized photomicrograph (...), the observation that "grains with c-axes oriented perpendicular to the plane of the section are dark in panels (B) and (D)" cannot strictly be upheld. Whether or not a grain appears dark on a cross- polarized photomicrograph not only depends on the inclination of the quartz c-axis w/ r to the image plane, but also on the orientation w/r to the horizontal and vertical direction within the image plane. In fact you also find dark grains in Fig. 4d that are not dark in Fig. 4d. If you had used circular polarization, the correlation between Fig. 4b and 4d would be perfect.Please modify or delete this remark from the caption.

Thanks for the information on circular polarizers. I'd like to get one. We rephrased it in a way that is actually true: "The blue spectrum peak (~470 nm, 2.65 eV) is dependent on quartz c-axis orientation, since dark grains in panel (B) are also dark in panel (D)."

Page 22 - line 3 - Figure 5 - "The data are jittered along the x axis to increase visibility." As you are plotting Ti concentrations in quartz vs. the distance, this sentence makes no sense here. Please delete.

>This comment was also sent in a follow up email:

After I pressed the submit button, I suddenly realized that I had misunderstood something w/r to your your Figures 5 7 and 9 (you'll see when you read my comments). In the captions of those figures, you wrote that the data are jittered. I could not understand why you would want to mention this - I thought you could just see it. I thought that you had plotted the data at distances measured in the field.. And therefore I wrote in my review that this comment was unnecessary Now it occurs to me that maybe the distances of the data points are not measured (or calculated) actual distances, but approximate distances typical of 4 or so separate horizons?

So instead of removing the sentence you could maybe clarify that the jittering is introduced "artificially"...? (Maybe I am not the only one to get it wrong...)

Most of the data (>90% I think), is from one transect. There are  $\sim$ 15 locations along this transect, so a lot of the data would plot on top of each other at the different localities. What we have done is shift the plotted distance values from their true values by adding some random noise (jittering) to the measured distance values. Without doing this, the trends and concentrations of data can't be seen well. Since this wasn't clear, we've replaced the sentence with "The data are jittered along the x axis by an amount not exceeding 30 km in order to increase the visibility of the data." Hopefully this is clearer.

Page 22 - line 11 Figure 6.No, you are not plotting "grain size vs. Ti concentration" but the reverse. One always plots Y against X (where X is the independent variable and usually the horizontal), Please correct.

This has been corrected.

Page 23 Figure 7 (cf. Figure 4). Why did you use uppercase A,B while references to image are lowercase? I am not sure if Solid Earth has a policy about using upper- or lowercase.

This is strange. In the version we have and the most recent one online (se-2018-12-manuscript-version2.pdf), we didn't do that. The figure captions use upper case references, e.g. "Data from Cross et al. (2015) are plotted in panel A but are not available for panel B" and "The lack of a clear trend in (B)"

Page 23 - line 13 Figure 7 (cf. Figure 5)"The data are jittered along the x axis to increase

visibility."Again, as you are ostensibly plotting Ti concentrations in quartz vs. the distance, this sentence makes no sense here.Please delete

We've changed it to "The data are jittered along the x axis in order to increase visibility."

Page 24 - line 11 Figure 8 Again, you are not plotting "grain size vs. Ti concentration" but the reverse. (see Fig. 6)Please correct.

This has been corrected.

Page 25 - line 11Figure 9"Data are jittered along the x-axis."Same as in Figure caption 7, this sentence makes no sense here. Please delete.Instead, please comment on the inset.

Now changed to "The data are jittered along the x axis by an amount not exceeding 50 km in order to increase visibility."

Page 26 - line 4 Figure 10 "Image" ? Please specify.

Replaced "image" with "photomicrograph"

Comments (respond at your own discretion):

Page 1 line 31 - you say "...where the examination of plate boundary phenomena ... can be informed by observation ...."getting close to Yogi Berra's famous quote: "You can observe a lot by just watching." :-)

True. We changed it to "the understanding of..." rather than "examination of"

Page 2 - line 30 - you say: "classic sequence of fault rocks: protomylonites, mylonites, ultramylonites, and finally cataclasites" - cataclasites are "classical" if the fault development is following a cooling path or later-stage brittle overprint....

Unchanged

Page 3 - line 3 - your write: "clockwise rotation (when viewed from the SW) and dextral shearing..." - I am not familiar with the details of the Alpine fault deformational history... So I wonder: are "clockwise rotation" and "dextral shearing" two different events or both the same? Specifying the viewing direction ("when viewed from the SW") indicates to me that the "rotation"

must be on a vertical or very steep plane, and presumably applies to the \*dextral shearing" too. If so, and if both "clockwise rotation" and "dextral shearing" are the same you might consider calling this " top to the NE shearing" and save yourself the viewing direction...

They are considered to be different things.

Page 3 - line 10 - What is the basis for the strain determinations? Any references ?

Added "based on thickness of offset, mylonitized pegmatite veins (Norris and Cooper, 2003)"

Page 3 - line 38 (Page 4 - line 1) - why "common in either" - why not "common in both" or simply "common in" ?

No good reason. We removed the word "either"

Page 23 - line 10-13 - Figure 7 - Very long caption. "The lack of a clear trend in (B) suggests that..." etc. until "... not available for panel B."You may consider inserting (and possibly shortening) this part of the caption into the running text on page 6 - line 19.

We removed the sentence about panel B.

Page 24 - line 5-11 Figure 8 Very long caption. "Away from the fault this trend..." etc. until "... of fine grains in the ultramylonite." I think it would be better to move this descriptive part of the caption to the running text on page 6 - line 35.

We agree it's too long. There was also some duplicated information even within the caption. We shortened and edited it to "Figure 8. Ti vs. grain size for Gaunt Creek samples. Samples at structural distances <160 m from the fault are shown in black (mainly ultramylonites), while mylonites and protomylonites are plotted in transparent purple. The trend in the samples close to the fault suggests late recrystallization (as indicated by finer grain size) occurred at conditions where lower concentrations of Ti were stable. Only partial equilibration of Ti-in-quartz values was achieved during recrystallization of fine grains in the ultramylonite. Away from the fault there is not a general trend of fine grains having lower Ti concentrations."

Page 27 - line 5-10 Figure 11Very long caption. "Ti activity is constrained by the fact that..." etc. until "... as summarized by Toy et al. (2010)."Consider moving this part of the caption to the running text on page 8 - line 9.

Agreed that it's too long. We removed the sentence "The geothermal gradient proposed here is

based on new data indicating mylonite and protomylonite deformation occurred at temperatures from roughly 600 to 450 °C whilst Ti-in-quartz concentrations of ~3 ppm were stable" since this is already clearly stated elsewhere.

Page 28 - Figure 12 In a time where it has become fashionable to attribute every brittle microstructure to an earth quake, I would welcome if you could label the horizontal line in Figure 12 as "limit of seismicity" (as in text) and remove "Earthquakes" and "No Earthquakes". - Else, the next thing you know is somebody citing your paper as a demonstration of earthquakes as producing ultramylonites with low Ti-in-quartz values.

OK, we've made this change.

[revised manuscript text omitted]

---

## Author Response (AR3)

9/4/2018

Hi Renée,

Thanks for your help with the paper on the Alpine Fault Ti-in-quartz. Just a quick point of clarification in response to your recent message—the "grain sizes" that we are mainly discussing in the paper are the sizes of particular grains that were also analyzed by SIMS. So there wasn't any averaging involved (means, modes, etc.). We only referred to average grain size a couple times but either cited earlier literature for those estimates or gave broad ranges, e.g. "has a grain size of 20-200 $\mu$m", or "grain size on the order of 10-20 $\mu$m."

I have added "manually" as suggested though, as that should clarify what we did.

I believe this concludes the changes and the enclosed manuscript should be ready for type-setting.

Thank you for your editorial work,

Steve Kidder